# Are Medical Vision–Language Foundation Models Ready for Dermatology

## Abstract

Medical Vision-Language models (VLMs) show significant promise for clinical image understanding, offering the promise of greater medical accessibility and interpretability. However, a critical performance gap in diagnostic accuracy exists between their strong vision encoders and the full multimodal model. This performance gap suggests that such VLM fails to make full use of the strength of its vision branch. Such misalignment also implies that these models often over-rely on their language priors, producing plausible-sounding diagnoses without sufficiently grounding their reasoning in visual evidence. Focusing on dermatology, we systematically investigate the root causes of this phenomenon. While fine-tuning can improve accuracy, it often compromises the model's essential reasoning capabilities. To address these challenges, we introduce a inference-time pipeline designed to close the performance gap while preserving the model's reasoning abilities. Our pipeline enhances diagnostic accuracy and faithfulness without requiring additional training. These strategies are readily extensible, suggesting a path toward more reliable and interpretable VLMs in medicine and beyond.

## 1 Introduction

Diagnosing conditions from dermatological images is a challenging task due to inherent complexities, such as subtle variations in disease presentation and a lack of standardized image quality. While advances in computer vision have enabled diagnostic models to achieve expert-level accuracy (Brinker et al., 2019; Esteva et al., 2017; Liu et al., 2020; Soenksen et al., 2021; Wang et al., 2024), their deployment in safety-critical domains like healthcare demands more than just precision. It is crucial that a model's decisions are interpretable, ensuring they are grounded in clinical evidence rather than spurious correlations. The recent emergence of Vision-Language Models (VLMs) in dermatology addresses this need, aiming not only for accurate diagnosis but also for explainable reasoning, thereby paving the way for the reliable, real-world application of AI in medicine.

The development of dermatological VLMs has often relied on pre-training with large-scale, specialized datasets and advanced architectures. For example, DermLIP is a CLIP-based VLM (Radford et al., 2021) that was pre-trained on Derm1M, a dataset with over one million image-text pairs, enabling tasks like zero-shot classification and concept identification (Yan et al., 2025a). Similarly, MONET (Kim et al., 2024) has fine-tuned a CLIP model on over 100K dermatological images paired with natural language descriptions from a large collection of medical literature, aiming for interpretable diagnoses. Additionally, SkinVL (Zeng et al., 2025) and SkinGPT-4 (Zhou et al., 2024) have integrated large language models into their dermatological VLMs and leveraged large dermatological corpora to facilitate classification with nuanced disease interpretation and visual question answering. More recently, MedGemma, a family of medical foundation models, has emerged to demonstrate superior medical reasoning capabilities (Sellergren et al., 2025). Built upon the powerful Gemma 3 (Team et al., 2025) architecture and incorporating a medically tuned vision encoder MedSigLIP, MedGemma excels at transparent reasoning, making it an ideal for clinical applications.

In this work, we uncover a critical paradox: while MedGemma's vision encoder, MedSigLIP, possesses exceptional discriminative power, this strength fails to translate into the zero-shot diagnostic accuracy of the full multimodal model. As shown in Fig. 1, we observe that a simple few-shot linear probe on MedGemma's vision encoder, which is medically tuned from SigLIP (Zhai et al.,

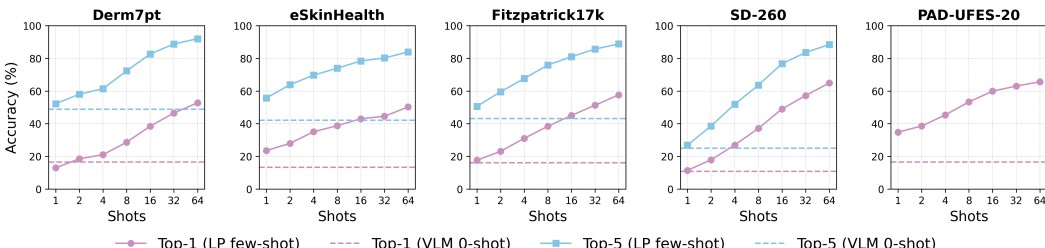

Figure 1: A comparison of the zero-shot performance of the MedGemma-4B VLM against the linear probe performance of its vision encoder, MedSigLIP. In this figure, LP denotes a linear probe on the vision encoder, while VLM denotes a direct query for diagnosis to the full vision-language model.

Table 1: Classification performance across five dermatology datasets. The table compares the performance for MedGemma-4B under three conditions: zero-shot inference, full-data fine-tuning, and a linear probe on its vision encoder (MedSigLIP). Performance is compared against a linear probe on a competitive vision encoder, PanDerm. Linear probing on both vision encoders significantly outperforms the zero-shot with the VLM, even after fine-tuning. Results are averaged over five runs.

| Data Details | | | | MedGemma-4B | | | | | | PanDerm | |
|---|---|---|---|---|---|---|---|---|---|---|---|
| | | | | Zero-Shot | | Fine-tune | | Linear Probe | | Linear Probe | |
| Datasets | #Class | #Train | #Test | ACC | F1 | ACC | F1 | ACC | F1 | ACC | F1 |
| Derm7pt | 14 | 413 | 395 | 16.46 ($\pm$ 0.02) | 17.77 | 39.50 ($\pm$ 2.15) | 22.49 | 57.22 ($\pm$ 0.88) | 30.07 | 58.99 ($\pm$ 0.76) | 57.93 |
| eSkinHealth | 24 | 2,714 | 2,676 | 13.35 ($\pm$ 0.15) | 16.33 | 60.26 ($\pm$ 1.42) | 56.17 | 65.35 ($\pm$ 0.54) | 42.51 | 60.64 ($\pm$ 0.61) | 59.83 |
| Fitzpatrick17k | 20 | 3,100 | 3,100 | 16.73 ($\pm$ 0.08) | 11.13 | 47.32 ($\pm$ 1.10) | 45.16 | 66.32 ($\pm$ 0.45) | 66.90 | 64.87 ($\pm$ 0.39) | 64.88 |
| SD-260 | 260 | 10,362 | 10,238 | 10.81 ($\pm$ 0.05) | 7.94 | 39.45 ($\pm$ 0.95) | 34.63 | 75.35 ($\pm$ 0.32) | 60.10 | 72.08 ($\pm$ 0.28) | 60.24 |
| PAD-UFES-20 | 6 | 1,134 | 1,164 | 46.91 ($\pm$ 0.12) | 32.98 | 66.75 ($\pm$ 1.85) | 61.53 | 76.03 ($\pm$ 0.91) | 66.22 | 74.21 ($\pm$ 0.84) | 64.98 |

2023a), achieves remarkable classification accuracy across five skin disease datasets. Direct comparisons reveal that MedSigLIP is comparable to, and at times superior to, PanDerm (Yan et al., 2025b), the specialized dermatology vision encoder of DermLIP, as shown in Table 1. Taken together, these results confirm that MedGemma's vision branch possesses powerful discriminative capabilities that are underutilized by the full multimodal model's zero-shot classification performance. Although fine-tuning may seem promising, our results indicate that applying LoRA (Hu et al., 2022) to MedGemma underperforms a linear probe and often compromises the model's reasoning abilities—the very reason for using a foundation model in the first place (see Table 1).

This discrepancy leads us to two fundamental questions: **(1)** What are the root causes of the large performance gap between MedGemma's highly capable vision encoder and its end-to-end zero-shot diagnostic performance? **(2)** Can we develop fine-tuning-free strategies to close this gap, improving zero-shot accuracy while preserving the model's essential reasoning abilities? To answer these questions, our work makes the following contributions:

- We provide a systematic analysis of MedGemma-4B, identifying and verifying three root causes of the performance gap: training data distribution mismatch, under-reliance on the vision branch, and a fundamental misalignment between the discriminative objective of the encoder and the generative objective of the language model.

- We introduce a fine-tuning-free inference pipeline that guides the VLM to enhance zero-shot diagnostic accuracy and faithfulness without any model updates.

- We offer a set of actionable guidelines for practitioners to deploy large, often "black-box," VLMs more effectively, maximizing their performance in resource-constrained settings.

While our investigation is grounded in dermatology using MedGemma-4B, the identified challenges and proposed solutions extend far beyond our specific use case. The over-reliance on language priors and modality misalignment are fundamental issues facing the broader field of vision-language research (Zhai et al., 2023b; Yang et al., 2025; Hu et al., 2024; Tong et al., 2024). Therefore, our analysis provides valuable insights into a generalizable approach to improving the reliability and visual grounding of VLMs in safety-critical applications beyond dermatology or healthcare.

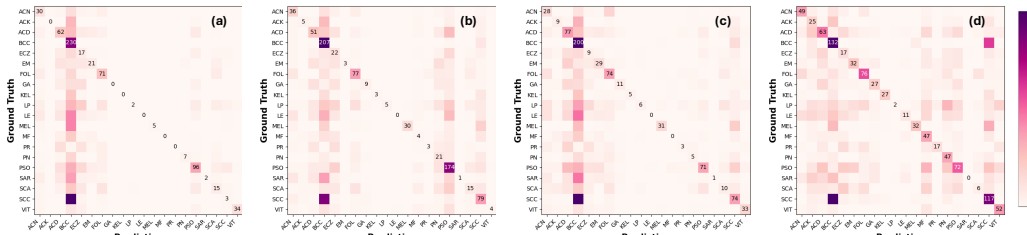

Figure 2: These confusion matrices compare the impact of each fine-tuning-free strategy on the Fitzpatrick17k dataset: (a) "direct answer", (b) "in-context", (c) a "describe-then-decide", and (d) a "top-5 to top-1" following a 2-shot linear probe. For class abbreviations, see Appendix 11.

## 2 PRELIMINARY

The MedGemma family of multimodal models is available in two variants: a 4B and a 27B parameter version. Both models employ the MedSigLIP image encoder, which was pre-trained on a diverse corpus of de-identified medical data, including chest X-rays, dermatology images, ophthalmology images, and histopathology slides. The primary distinction between the two is that the 27B variant was also pre-trained on an additional medical corpus that includes electronic health records (EHRs). As our research focuses on visual diagnosis from images rather than EHR data, MedGemma-4B is a more suitable choice for this study. We formalize the interaction with the VLM $\mathcal{M}$, as follows:

$$\mathcal{M}[\text{Prompt}, \text{Image}] \rightarrow \text{Response} , \tag{1}$$

where the model receives a textual prompt and an image of a skin condition as inputs and generates a response containing a diagnosis and an explanation.

**Datasets.** To ensure a comprehensive and robust evaluation, our study considers five distinct datasets selected to cover a wide spectrum of diseases and patient demographics. For broad coverage, we include SD-260 (Yang et al., 2019), which contains clinical images across 260 skin conditions, and Fitzpatrick17k (Groh et al., 2021), which features light-skinned images annotated with Fitzpatrick Skin Types. Following the methodology of (Wang et al., 2025b) to leverage their disease checklists, we use the same subset of Fitzpatrick17k as that used in their work. To assess performance on more specific diseases and diverse populations, we incorporate Derm7pt (Kawahara et al., 2019), focusing on its clinical photos of skin cancer, and eSkinHealth (Wang et al., 2025a), a specialized dataset of Neglected Tropical Diseases (NTDs) in West African populations. Finally, to test the model's performance on familiar data, we also include PAD-UFES-20 (Pacheco et al., 2020), an in-domain dataset of skin lesions that was used during MedGemma's pre-training. For datasets that do not provide an official train-test split, we partitioned the data by case and class with a 0.5 split ratio, with further details provided in Table 1 and Appendix. 12. Implementation details and hyperparameter tuning results for fine-tuning and linear probing are provided in the Appendix. C.

## 3 METHODS

### 3.1 HYPOTHESIS 1: TRAIN-TEST DISTRIBUTION MISMATCH

We first hypothesize that the performance gap stems from a train-test distribution mismatch. That is, the model may not have learned sufficient representations for certain conditions if they were rare or absent in its training data. This distribution mismatch prevents the model from generalizing to these "tail" or out-of-distribution classes during zero-shot inference.

**Evidence.** While the exact composition of MedGemma's training data is not public, we can investigate this hypothesis by analyzing the model's prediction bias. The confusion matrix in Fig. 2 clearly demonstrates this bias: the model shows a strong tendency to over-predict certain common conditions, such as basal cell carcinoma (BCC). For instance, nearly all cases of lupus erythematosus (LE) are misclassified as BCC. This raises a critical question: is this error due to a bias in the visual pre-training data (a vision branch issue) or from an incomplete conceptual understanding in the language model (a language branch issue)? By leveraging the model's reasoning capabilities, we can

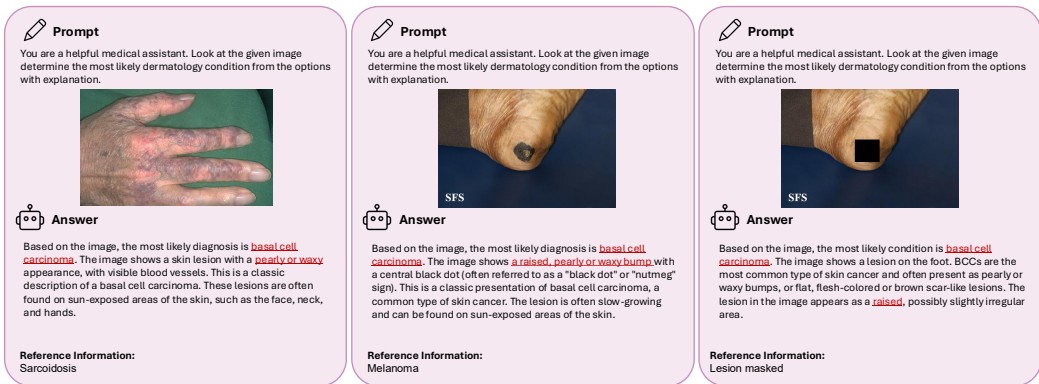

Figure 3: Examples of incorrect answers and hallucinated explanations from MedGemma-4B. The model incorrectly diagnoses sarcoidosis (left) and melanoma (middle) as basal cell carcinoma, providing descriptions of features that are not present in the images. In the third example (right), the lesion is manually masked, yet the model generates the same incorrect diagnosis and explanation, indicating it may generate a prediction first and rationalize it with a post-hoc explanation.

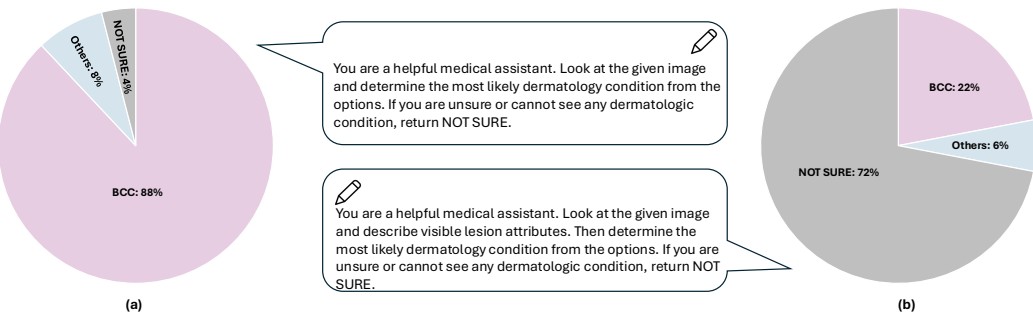

Figure 4: Abstention Test. The model is tested on 200 images with masked lesions. In (a), we present the images and prompt the model to **diagnose** based on visible signals. In (b), we present the same images and prompt the model to first **describe** the visible lesion before diagnosing. Note that the "NOT SURE" option was exclusively used for this abstention test, and was not included in the options list of the standard forced-choice classification task.

probe its conceptual knowledge. We prompted the model to describe the characteristics of diseases it consistently fails to diagnose (i.e., those with 0% zero-shot accuracy). As shown in Appendix.14 and Appendix 15, the model is able to generate equally detailed and accurate descriptions for BCC and LE, suggesting its language-based understanding of these rare diseases is intact. This evidence indicates the problem most likely arises from an imbalanced visual training distribution.

**Solution.** Although the model may lack sufficient visual examples of rare diseases, it possesses a foundational knowledge of general dermatological terms (e.g., "papules", "inflammation," "reddish," ). We can leverage this by re-framing the task from recognizing a disease name to matching visual evidence with a clinical description. Following the methodology of (Wang et al., 2025b), our solution provides the model with contextual information in the prompt. Alongside each potential disease name in the list of options, we include a description of its key visual characteristics using a set of generic dermatological vocabularies. Instead of forcing the model to map visual cues to a potentially unfamiliar disease label, this approach encourages it to map visual cues to a provided description. As shown in the improved confusion matrix in Fig. 2 (b), this strategy effectively guides the model toward more accurate and diverse predictions. See visual descriptions in Appendix. 13.

## 3.2 HYPOTHESIS 2: UNDER-RELIANCE ON THE VISION BRANCH

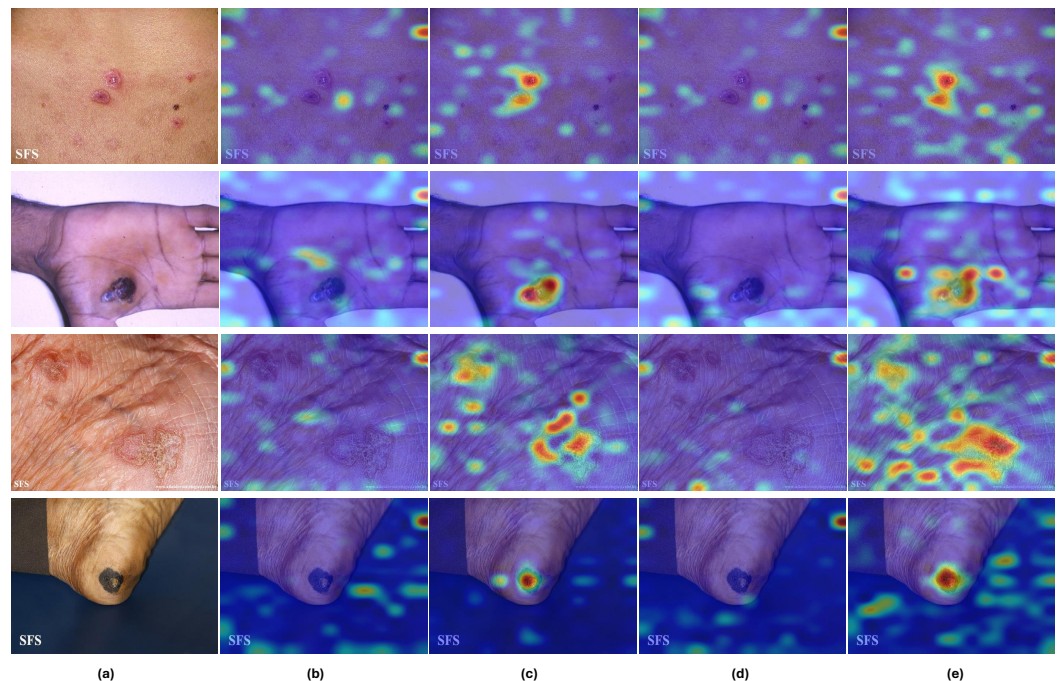

(a)     (b)     (c)     (d)     (e)

Figure 5: Attention heatmaps. Columns show (a) the original image, (b, d) attention from the prompt querying direct answer (baseline), and (c, e) attention from our "describe-then-decide" strategy. We visualize attention for the final diagnosis tokens (b, c) and for the entire response (d, e). The 'describe-then-decide' prompt yields attention maps that are more accurately concentrated on the lesion area, both during the final decision (c) and across the full response (e), confirming improved visual grounding.

We hypothesize that the VLM's powerful, heavyweight language model component leads it to overrely on textual priors when making a diagnosis. This behavior, documented in prior works (Yang et al., 2025; Tong et al., 2024), causes the model to generate a plausible-sounding diagnosis without sufficiently grounding its prediction and reasoning in the visual evidence of the image.

**Evidence.** We first observe that the model is prone to hallucinated reasoning, generating explanations that are not grounded in the image. As shown in Fig. 3, when presented with an image of "sarcoidosis", MedGemma incorrectly identifies the condition as BCC and provides a factually incorrect explanation, as there is no visual evidence of a "pearly or waxy" lesion. Similar observations are made for an image of melanoma, even when the lesion is intentionally masked. In these cases, the model tends to provide a

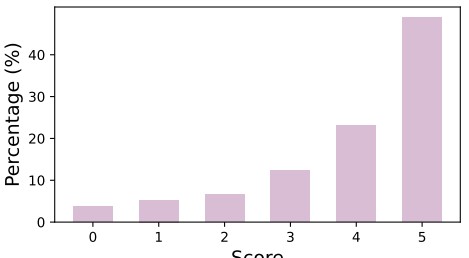

Figure 6: Distribution of quality scores from our human evaluation study. A dermatologist rated 100 generated descriptions, with the score [0-5] indicating the number of clinical criteria correctly described. The results indicate a high level of quality and accuracy.

generic, post hoc explanation for BCC, indicating that its diagnosis is not derived from the image's visual features. To quantify this behavior, we masked 200 images and tested whether the model would abstain from making a diagnosis when the primary lesion was hidden. As shown in Fig. 4(a), the model failed to abstain in 88% of cases, instead defaulting to a BCC diagnosis. This provides evidence that the model makes predictions without using its "eyes."

**Solution.** To mitigate this over-reliance on language priors, we introduce a "describe-then-decide" prompting strategy. This method forces the model to first articulate the visible features of the lesion

before making a final diagnosis, thereby compelling it to "look at" the image. The premise is that an accurate diagnosis should be preceded by an accurate visual description. This simple intervention yields improved classification accuracy, as shown in the enhanced confusion matrix in Fig. 2(c) and Table. 4. Additionally, when presented again the 200 masked images, the model's abstention rate increased from 12% to 72%, evidencing that this prompt encourages visual grounding.

To thoroughly validate that our "describe-then-decide" strategy effectively focuses the model on the lesion area, we generate and analyze attention heatmaps. Here, we define two metrics:

**Label-Token Heatmap** ($\mathbf{m}_{\text{label}}$): This measures the average attention the model pays to the input image tokens specifically when generating the words of the final diagnosis (e.g., "basal cell carcinoma"). It helps us see what the model attends to at the moment of decision.

**All-Tokens Heatmap** ($\mathbf{m}_{\text{all}}$): This measures the average attention paid to the image tokens across the entire generated response, including both the description and the diagnosis.

Specifically, let the generated sequence have $S$ tokens. At step $t$, after averaging head layers, the decoder gives an attention row $\bar{\mathbf{a}}^{(t)} \in \mathbb{R}^{T_t}$. Let the image occupy a contiguous token block $\mathcal{I} = \{s_{\text{img}}, \dots, s_{\text{img}} + L_{\text{img}} - 1\}$. Define the image-restricted vector, the slice of a token's attention only over the image tokens, with everything else (text tokens) dropped, as

$$\mathbf{r}^{(t)} \in \mathbb{R}^{L_{\text{img}}}, \qquad \mathbf{r}^{(t)}[j] = \begin{cases} \bar{a}^{(t)}_{s_{\text{img}}+j}, & s_{\text{img}} + j \leq T_t, \\ 0, & \text{otherwise.} \end{cases} \tag{2}$$

So $\mathbb{R}^{L_{\text{img}}}$ is just a length-$L_{\text{img}}$ real vector—one weight per visual token. To check how much the model attends to the final lesion prediction when generating responses (see Fig. 5 (b-c)), the label-token heat vector is

$$\mathbf{m}_{\text{label}} = \frac{1}{|\mathcal{S}_{\text{label}}|} \sum_{t \in \mathcal{S}_{\text{label}}} \mathbf{r}^{(t)} \in \mathbb{R}^{L_{\text{img}}} \tag{3}$$

where $\mathcal{S}_{\text{label}}$ indexes the tokens that spell the predicted diagnosis. Reshape $\mathbf{m}_{\text{label}}$ to the token grid and upsample to the image to render the heatmap. For the entire-response attention maps (see Fig. 5 (d-e)), all-tokens heat vector is

$$\mathbf{m}_{\text{all}} = \frac{1}{S} \sum_{t=1}^{S} \mathbf{r}^{(t)} \in \mathbb{R}^{L_{\text{img}}} \tag{4}$$

By comparing the heatmaps from a baseline prompt(that queries direct answer, 5 (b,d)) with those from our "describe-then-decide" prompt (Fig. 5 (c,e), we can isolate the effect of our strategy. When the label-token heatmap $\mathbf{m}_{\text{label}}$ is more concentrated on the lesion, it indicates that requiring a description forces the model to re-consult the visual evidence before making its final choice. Similarly, the "describe-then-decide" prompt yields broader yet meaningful attention in the $\mathbf{m}_{\text{all}}$ heatmap, as the model must scan the image to enumerate pixel-level attributes for the description.

### 3.3 HYPOTHESIS 3: MISALIGNMENT BETWEEN THE LANGUAGE AND VISION BRANCHES

While "describe-then-decide" improves visual grounding, it does not fully close the performance gap. This leads to our third hypothesis: a fundamental objective misalignment between the VLM's components. The vision encoder is optimized for discriminative tasks (distinguishing classes), whereas the large language model is optimized for generative tasks (explaining and reasoning). We hypothesize that the VLM is better suited for explanation and description than for forced-choice classification and that leveraging these distinct strengths is key to maximizing performance.

**Evidence.** Previous sections have already established that the vision encoder's discriminative performance surpasses that of the VLM. To complete our evidence, we now evaluate the VLM's primary strength: its ability to generate high-quality, descriptive text. Following the methodology of (Wang et al., 2025b), we prompted MedGemma-4B to describe lesions using a structured format covering five key clinical criteria:[Location Site, Lesion Type, Shape/Border, Color, Texture]. An example of a generated description is shown in Fig. 7. To quantitatively assess quality, was instructed to

**Prompt**

You are a helpful medical assistant. Look at the given image and describe the visible skin condition in terms of location, lesion type, shape, color and texture.

**Answer**

**Location:** The lesion is located on the dorsum (back) of the hand, specifically on the fingers.
**Lesion Type:** The lesion appears to be a combination of several different types, including:
   * Telangiectasias: These are small, dilated blood vessels that appear as fine, red lines or streaks.
   * Hyperkeratosis: This refers to thickening of the outer layer of skin (epidermis). It can appear as rough, scaly patches.
   * Possible Erythema: There is some redness present, which could indicate inflammation or increased blood flow.
**Shape:** The lesion is irregular in shape, with a somewhat patchy distribution across the fingers. It doesn't have a distinct border.
**Color:** The lesion exhibits a range of colors, including:
   * Red/Pink: This is due to the telangiectasias and possible erythema.
   * Brownish/Yellowish: This could be due to hyperkeratosis or changes in the underlying skin.
**Texture:** The texture is rough and scaly, consistent with hyperkeratosis. The telangiectasias may also contribute to a slightly rough feel.
**Overall Impression:** The lesion is a complex one, likely representing a combination of vascular and epidermal changes. The presence of telangiectasias, hyperkeratosis, and possible erythema suggests a possible underlying condition.

Figure 7: An example of a description generated by MedGemma-4B. When prompted to describe the lesion from Fig.3 instead of diagnosing it, the model produces a detailed and clinically relevant analysis of its features. This faithful description contrasts sharply with the hallucinated reasoning.

Table 2: Effectiveness of the proposed two-stage inference pipeline. Comparison of the **Baseline VLM** (MedGemma zero-shot), the standalone **Linear Probe** (MedSigLIP), and **Our Pipeline** which uses the Linear Probe to filter candidates (Top-5) for the VLM (Top-1). All accuracy values are Top-1 (%). Ablation study on different number of shots can be found in Appendix C.3

| Dataset | Baseline VLM *Zero-Shot* | Linear Probe *(Vision Encoder Only)* | | Our Pipeline *(LP Top-5 → VLM Top-1)* | |
|---|---|---|---|---|---|
| | **0-shot** | **1-shot** | **8-shot** | **1-shot** | **8-shot** |
| Derm7pt | 16.46 ($\pm$ 0.01) | 12.96 ($\pm$ 1.75) | 28.66 ($\pm$ 1.38) | 20.02 ($\pm$ 2.39) | 35.56 ($\pm$ 0.99) |
| eSkinHealth | 13.35 ($\pm$ 0.23) | 23.51 ($\pm$ 1.91) | 38.80 ($\pm$ 1.06) | 30.98 ($\pm$ 2.85) | 42.05 ($\pm$ 1.48) |
| Fitzpatrick17k | 16.73 ($\pm$ 0.19) | 17.63 ($\pm$ 3.13) | 38.43 ($\pm$ 1.03) | 25.35 ($\pm$ 2.41) | 37.82 ($\pm$ 1.37) |
| SD-260 | 10.81 ($\pm$ 0.13) | 11.48 ($\pm$ 2.27) | 37.12 ($\pm$ 1.46) | 22.31 ($\pm$ 1.95) | 42.33 ($\pm$ 0.92) |
| PAD-UFES-20 | 46.91 ($\pm$ 0.20) | 34.85 ($\pm$ 1.83) | 53.33 ($\pm$ 1.34) | 58.33 ($\pm$ 2.24) | 80.31 ($\pm$ 1.23) |

review the generated text against the original image and verify the accuracy of each of the five requested criteria. The resulting quality score (0-5) strictly reflects the number of criteria correctly described. As shown in Fig. 6, the model consistently achieves high scores (predominantly 4 or 5), confirming that it can accurately perceive and articulate visual features. This performance stands in stark contrast to the hallucinated and visually ungrounded explanations observed during the standard zero-shot diagnosis (see Fig. 3), providing strong evidence that the model possesses the necessary visual information but fails to effectively utilize it for the specific task of classification.

**Solution.** Our solution is a two-stage pipeline that leverages the complementary strengths of the vision and language components. Instead of forcing the VLM to perform a task it is not optimized for, we delegate the initial discriminative work to the vision encoder. First, we use the powerful vision encoder with a few-shot linear probe to identify the top-5 most likely diagnosis candidates for a given image. This narrows the field to a small set of high-probability candidates. Then, we feed these top-5 candidates to the full VLM and prompt it to make the final diagnosis from only that reduced set. We term this strategy as "Top-5 to Top-1".

As shown in Table. 2, this approach is highly effective, especially in low-data regimes. With 1 to 8 shots, the two-stage pipeline's accuracy is significantly higher than that of a linear probe alone. The benefit diminishes as the number of shots increases (e.g., 16 or more), as the linear probe becomes powerful enough on its own (see Appendix C.3). This makes our approach particularly valuable for real-world medical applications where labeled data is scarce.

This strategy offers three additional benefits. First, by providing the VLM with only 5 options instead of 20 or more, it reduces the prompt length, saving token space for more context. Second, the vision encoder is a plug-and-play component; practitioners can substitute MedSigLIP with any

Table 3: Zero-shot prompts.

| Type | Prompt |
|------|--------|
| direct answer (baseline) | You are a helpful medical assistant. Look at the given image and determine the most likely dermatology condition from the options:
(A) Granuloma annulare
(B) Lupus erythematosus
(C) Vitiligo.
... ... |
| in-context | You are a helpful medical assistant. Look at the given image and determine the most likely dermatology condition from the options:
(A) Granuloma annulare, characterized by smooth skin-colored or pink papules arranged in an annular ring with firm ridge.
(B) Lupus erythematosus, characterized by scaly plaques with follicular plugging and central scarring on sun-exposed skin; acute malar rash appears as flat red butterfly across cheeks sparing folds.
(C) Vitiligo, characterized by sharply bordered milky white macules on face, hands, or genital skin with normal texture and pigment loss.
... ... |
| describe-then-decide | You are a helpful medical assistant. Look at the given image and describe visible lesion attributes (location, lesion type, shape/border, color, texture). Then determine the most likely diagnosis from the options:
(A) Granuloma annulare
(B) Lupus erythematosus
(C) Vitiligo.
... ... |

other powerful, domain-specific vision encoder to potentially boost performance further. Third, and perhaps most importantly, we still have the model's reasoning capabilities at our disposal.

## 3.4 ABLATION STUDY AND COMBINED PERFORMANCE

Having investigated the root causes of the performance gap and proposed a corresponding remedy for each, we now conduct an ablation study to evaluate the individual and combined effects of our strategies. We compare the following five zero-shot prompting configurations, with the results presented in Table. 4. For the "Top-5 to Top-1" and "all combined" configurations, we use a 8-shot linear probe to generate the initial candidates. We choose 8 shots because, as shown in Table 10, it represents a point of high efficacy in the low-data regime before performance gains begin to saturate. For the SD-260 dataset, providing expert-verified clinical descriptions for all 260 classes was infeasible. Therefore, for this study, we use a subset of SD-260 containing only the classes that overlap with the other datasets. Example of the prompts are shown in Table. 3.

The results of the ablation study provides evidence for the effectiveness of the proposed fine-tuning-free pipeline. We observe that each of the three strategies (in-context, "describe-then-decide", and "top-5 to top-1") independently yields a significant improvement in Top-1 accuracy over the direct answer baseline across all five datasets. This validates our three-part analysis, confirming that each hypothesis addresses a distinct limitation of the VLM. Most importantly, the "all combined" design consistently achieves the highest performance, substantially outperforming any single strategy. This demonstrates that the benefits of each strategy are not only independent but also cumulative, working together to progressively close the performance gap and enhance diagnostic accuracy.

## 3.5 GENERALIZATION TO OTHER VLMS FOR DERMATOLOGY

To demonstrate that the identified challenges and proposed solutions are not specific to the MedGemma architecture, we extended our evaluation to SkinVL (Zeng et al., 2025). As a LLaVA-based model, SkinVL represents a distinct architectural lineage from MedGemma. First, we confirmed that the performance gap is a pervasive issue: consistent with MedGemma, a simple linear probe on SkinVL's vision encoder significantly outperforms its zero-shot VLM accuracy (see Ap-

Table 4: Ablation study on the effectiveness of each proposed strategy. This table presents the zero-shot Top-1 accuracy for five different configurations. The "direct answer" column serves as the baseline, using a standard classification prompt. The "in-context" strategy augments the prompt with clinical descriptions; "describe-first" requires the model to describe visual features before diagnosis; and "top-5 to top-1" is our two-stage strategy where the VLM reranks candidates from a 8-shot linear probe. The final "all combined" column integrates all proposed strategies.

| Dataset | direct answer | in-context | describe-first | top-5 to top-1 | all combined |
|---|---|---|---|---|---|
| Derm7pt | 16.46 ($\pm$ 0.17) | 26.72 ($\pm$ 0.82) | 30.98 ($\pm$ 1.37) | 35.10 ($\pm$ 1.95) | 38.27 ($\pm$ 0.81) |
| eSkinHealth | 13.35 ($\pm$ 0.23) | 30.07 ($\pm$ 1.51) | 29.77 ($\pm$ 0.92) | 43.94 ($\pm$ 1.03) | 48.00 ($\pm$ 0.86) |
| Fitzpatrick17k | 16.73 ($\pm$ 0.25) | 24.01 ($\pm$ 2.01) | 24.75 ($\pm$ 1.53) | 28.32 ($\pm$ 1.67) | 40.97 ($\pm$ 1.41) |
| SD-260 (subset) | 24.31 ($\pm$ 0.15) | 34.29 ($\pm$ 1.06) | 33.80 ($\pm$ 1.37) | 40.01 ($\pm$ 2.18) | 44.90 ($\pm$ 0.91) |
| PAD-UFES-20 | 46.22 ($\pm$ 0.17) | 52.23 ($\pm$ 1.26) | 62.33 ($\pm$ 1.58) | 75.23 ($\pm$ 2.07) | 84.68 ($\pm$ 1.80) |

pendix D for full comparison). Second, we evaluated our training-free pipeline on SkinVL. Due to its stricter context window, we adapted the pipeline to perform "Top-5 to Top-1" filtering first. As shown in Table 5, our strategies yield consistent performance gains across multiple datasets, even improving upon the high baselines of Fitzpatrick17k (a dataset included in SkinVL's pre-training). This confirms that our approach generalizes effectively to LLaVA-based architectures.

Table 5: Generalizability on SkinVL (LLaVA-based). Top-1 Accuracy (%) comparison. Our proposed inference pipeline consistently improves performance over the direct answer baseline, demonstrating robustness across different VLM architectures.

| Inference Strategy | Derm7pt | eSkinHealth | Fitzpatrick17k |
|---|---|---|---|
| Direct Answer (Baseline) | 12.77 ($\pm$ 0.12) | 23.76 ($\pm$ 0.21) | 10.05 ($\pm$ 0.08) |
| + Top-5 Filtering | 29.66 ($\pm$ 1.45) | 40.25 ($\pm$ 1.82) | 58.37 ($\pm$ 1.15) |
| + In-Context Description | 31.19 ($\pm$ 1.33) | 44.03 ($\pm$ 2.01) | 62.97 ($\pm$ 1.28) |
| **+ Describe-then-Decide (Ours)** | **32.56** ($\pm$ 0.95) | **44.89** ($\pm$ 1.67) | **63.91** ($\pm$ 1.05) |

# 4 RELATED WORKS

## 4.1 FOUNDATION MEDICAL VISION-LANGUAGE MODELS FOR DERMATOLOGY

Recent advancements in dermatology have been driven by a series of specialized VLMs (Zhang et al., 2024; Lin et al., 2023). DermLIP, for instance, was trained on Derm1M, a large-scale dataset of over one million image-text pairs, to enable zero-shot classification and concept identification (Yan et al., 2025a). Similarly, MONET was developed by fine-tuning a CLIP model on over 100,000 image-caption pairs from medical literature to achieve transparent, concept-based diagnoses (Kim et al., 2024). Such strategies of designing specialized curricula have also proven effective in other medical domains, such as retinal image analysis, to ensure robust feature learning for medical VLMs (Holland et al., 2025). Other models like. SkinVL (Zeng et al., 2025) and SkinGPT-4 (Zhou et al., 2024) have integrated large language models to facilitate more nuanced disease interpretation and visual question answering. SkinVL was trained on nearly 10,000 specialized image-captions and 27,000 QA pairs derived from professional textbooks, while SkinGPT-4 was aligned with Llama-2 (Touvron et al., 2023) using over 52,000 skin disease images. While these models have demonstrated strong performance on specific dermatological tasks, our work focuses on. MedGemma (Team et al., 2025; Sellergren et al., 2025), a generalist medical foundation model. We selected MedGemma because its architecture, featuring a powerful, medically-tuned vision encoder (MedSigLIP) combined with an advanced language model (Gemma 3), represents the state-of-the-art in transparent medical reasoning. This makes it an ideal testbed for investigating the fundamental vision-language alignment challenges that are broadly relevant to the next generation of medical AI.

## 4.2 Misalignment of VLMs and Over-reliance on Language Prior

A central challenge in current VLM is the misalignment between their powerful vision encoders and the end-to-end multimodal system (Xing et al., 2025). This often manifests as an over-reliance on the language model's priors, where the VLM fails to properly integrate visual information during reasoning. Studies have shown that as a VLM generates a response, its attention to visual input can gradually diminish, sometimes having a negligible influence on the final output (Yang et al., 2025). This tendency to "look away" from the image leads to models producing plausible-sounding but visually ungrounded responses and hallucinations. (Tong et al., 2024) investigated this issue, demonstrating that even advanced models like GPT-4V exhibit elementary visual shortcomings that can be traced back to weaknesses in the underlying CLIP vision encoder. They found that models struggle with basic visual patterns related to orientation, counting, and object attributes, suggesting the visual representations are not fully utilized. This misalignment is further exacerbated during instruction tuning, which can cause catastrophic forgetting; (Zhai et al., 2023b) found that fine-tuning often degrades the model's core visual perception, causing it to lose the robust classification abilities of its original vision encoder. To mitigate such inconsistencies, recent works have proposed automated structured reporting to standardize model outputs and reduce hallucination (Delbrouck et al., 2025). Taking together, these challenges underscore the critical need for strategies that enforce strong visual grounding in the final reasoning process of vision language models.

## 5 Conlusion and Dicussion

In this work, we investigated a critical paradox in medical foundation models: the significant performance gap between a VLM's powerful vision encoder and its end-to-end zero-shot diagnostic accuracy. Focusing on MedGemma-4B in the context of dermatology, we systematically identified and validated three root causes for this discrepancy: a train-test distribution mismatch, an over-reliance on language priors, and a fundamental objective misalignment between the vision and language components. To address these challenges, we introduced a fine-tuning-free inference pipeline composed of three distinct strategies: providing in-context clinical descriptions, enforcing a "describe-then-decide" reasoning process, and leveraging the vision encoder for candidate selection in a "Top-5 to Top-1" framework. Our experiments demonstrate that these interventions, both individually and combined, significantly improve zero-shot classification accuracy, reduce diagnostic bias, and enhance the model's visual grounding, all without requiring any costly fine-tuning that could compromise its valuable reasoning abilities. While our analysis centered on a single model in dermatology, the issues of modality misalignment and language-prior dominance are endemic to the broader field of vision-language research. Therefore, our proposed strategies offer a model-agnostic and data-efficient blueprint for improving the faithfulness and reliability of VLMs in any safety-critical domain where decisions must be grounded in tangible evidence. Future work should explore the application of these principles to other medical specialties and foundation models, paving the way for more robust and trustworthy AI in clinical practice.

ETHICS STATEMENT

We adhere to the ICLR Code of Ethics. All data are public and de-identified, and no protected health information is included. We do not expect our approach to introduce direct ethical risks or harmful societal outcomes.

REPRODUCIBILITY STATEMENT

We have taken care to make our results reproducible. The paper and appendix specify all datasets, the exact prompts and prompting variants used in every experiment (Table 3 and Appendix tables), the evaluation protocols and metrics, ablations, and analysis procedures (Section 3 and Table 4).

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

# **Appendix**

## A    USE OF LARGE LANGUAGE MODELS

We used a large language model (LLM) solely for language polishing—e.g., editing grammar, improving clarity and flow, and standardizing tone. The LLM did not contribute to research ideation, experimental design, analysis, or result generation. The LLM is not an author or contributor, and its use complies with ICLR's policy on LLM usage.

## B    LIMITATIONS

While our work provides valuable insights and effective remedies for the performance gap in medical VLMs, we acknowledge its limitations. The proposed fine-tuning-free strategies are designed to mitigate the identified issues and significantly improve zero-shot performance, but they do not represent a complete solution to the underlying architectural challenges of modality misalignment in foundation models. Therefore, future work should focus on extending this analysis to a broader range of models and medical domains. Validating these strategies on other modalities, such as radiology (X-rays), pathology, and ophthalmology, would be a critical next step to confirm their generalizability. Furthermore, exploring how these principles could inform the development of new VLM architectures, designed from the ground up to ensure better vision-language alignment, presents an exciting avenue for future research.

While we recognize that a full clinical utility study involving multiple experts and inter-rater reliability analysis is critical for real-world deployment, such an extensive study is beyond the scope of this technical analysis. A comprehensive clinical decision-making evaluation is a necessary and immediate priority for future work.

## C    IMPLEMENTATION DETAILS

### C.1    LoRA FINE-TUNING

To ensure a rigorous evaluation of the model's adaptability, we fine-tuned MedGemma-4B-it using Low-Rank Adaptation (LoRA). Given the high memory requirements of 4B parameter models, we employed QLoRA (Dettmers et al., 2023), loading the base model in 4-bit precision with bfloat16 compute dtype. We applied LoRA adapters to all linear layers to maximize model expressivity while keeping the number of trainable parameters low. We also explicitly set the embed_tokens module to be trainable. The optimization was performed using the fused AdamW optimizer with a linear learning rate scheduler and a warmup ratio of 0.03. To determine the optimal configuration, we conducted a grid search over the LoRA rank ($r$), scaling factor ($\alpha$), and learning rate. The search space and selected hyperparameters are detailed in Table C1. We trained for 10 epochs with a batch size of 4 and gradient accumulation steps of 4. The final selected configuration (Rank 8, Alpha 16, LR $1e^{-4}$) was chosen based on validation set performance and used for all reported fine-tuning results. Details can be found in Table. 6.

Table 6: Hyperparameter search space and selected configuration for LoRA fine-tuning.

| Hyperparameter | Search Space | Selected Value |
|---|---|---|
| LoRA Rank ($r$) | $\{4, 8, 16\}$ | 8 |
| LoRA Alpha ($\alpha$) | $\{8, 16, 32\}$ | 16 |
| Learning Rate | $\{5e^{-5}, 1e^{-4}, 2e^{-4}\}$ | $1e^{-4}$ |
| LoRA Dropout | - | 0.05 |
| Target Modules | - | All Linear Layers |
| Precision | - | 4-bit (QLoRA) |
| Optimizer | - | AdamW (Fused) |
| Training Epochs | - | 10 |

Table 7: Full grid search results for LoRA fine-tuning on the Fitzpatrick17k dataset. The optimal configuration selected for the final model is highlighted in bold.

| Rank ($r$) | Alpha ($\alpha$) | Learning Rate | Accuracy (%) |
|---|---|---|---|
| 4 | 8 | $5 \times 10^{-5}$ | 42.15 |
| 4 | 8 | $3 \times 10^{-4}$ | 46.12 |
| 4 | 8 | $4 \times 10^{-4}$ | 45.88 |
| 4 | 16 | $5 \times 10^{-5}$ | 41.80 |
| 4 | 16 | $3 \times 10^{-4}$ | 45.05 |
| 4 | 16 | $4 \times 10^{-4}$ | 44.20 |
| 4 | 32 | $5 \times 10^{-5}$ | 35.60 |
| 4 | 32 | $3 \times 10^{-4}$ | 39.40 |
| 4 | 32 | $4 \times 10^{-4}$ | 37.10 |
| 8 | 8 | $5 \times 10^{-5}$ | 43.50 |
| 8 | 8 | $3 \times 10^{-4}$ | 46.45 |
| 8 | 8 | $4 \times 10^{-4}$ | 46.10 |
| 8 | 16 | $5 \times 10^{-5}$ | 44.10 |
| 8 | 16 | $3 \times 10^{-4}$ | **47.32** |
| 8 | 16 | $4 \times 10^{-4}$ | 46.95 |
| 8 | 32 | $5 \times 10^{-5}$ | 38.90 |
| 8 | 32 | $3 \times 10^{-4}$ | 41.25 |
| 8 | 32 | $4 \times 10^{-4}$ | 40.50 |
| 16 | 8 | $5 \times 10^{-5}$ | 43.20 |
| 16 | 8 | $3 \times 10^{-4}$ | 45.90 |
| 16 | 8 | $4 \times 10^{-4}$ | 45.50 |
| 16 | 16 | $5 \times 10^{-5}$ | 44.80 |
| 16 | 16 | $3 \times 10^{-4}$ | 46.98 |
| 16 | 16 | $4 \times 10^{-4}$ | 46.40 |
| 16 | 32 | $5 \times 10^{-5}$ | 43.10 |
| 16 | 32 | $3 \times 10^{-4}$ | 46.15 |
| 16 | 32 | $4 \times 10^{-4}$ | 45.20 |

## C.2 LINEAR PROBE

For feature extraction, we utilized the pre-trained medgemma-4b-it vision encoder in bfloat16 precision. We extracted the global image representation by computing the mean of the last hidden state tokens rather than using a single [CLS] token. The head was trained using the AdamW optimizer and CrossEntropyLoss for 100 epochs with a batch size of 128, as shown in Table. 8.

Table 8: Implementation details and hyperparameters for Linear Probing experiments.

| Hyperparameter | Value |
|---|---|
| Vision Encoder | MedSigLIP (Frozen) |
| Feature Pooling | Global Average Pooling |
| Precision (Backbone) | `bfloat16` |
| Precision (Head) | `float32` |
| Optimizer | AdamW |
| Batch Size | 128 |
| Epochs | 100 |
| Learning Rate | $\{3 \times 10^{-5}, 3 \times 10^{-4}, 3 \times 10^{-3}\}$ |

## C.3 ABLATION STUDY ON THE NUMBER OF SHOTS

We selected the 8-shot setting based on the analysis presented in Table 10, where 8-shot maximizes the utility of our pipeline. At 8-shot, the vision encoder is strong enough to capture the correct diagnosis in the Top-5 (high recall), but not yet perfect at Top-1 (low precision), creating the ideal scenario for the VLM to apply its reasoning for re-ranking. Beyond 8 shots, the vision encoder becomes self-sufficient, and the added value of VLM reasoning diminishes.

## D PERFORMANCE ANALYSIS ON SKINVL

Table 9: Performance Gap Analysis on SkinVL. Comparison of the zero-shot VLM performance against full Fine-Tuning (FT) and a Linear Probe (LP) on the vision encoder. Consistent with MedGemma, the Linear Probe significantly outperforms the zero-shot VLM, confirming the under-utilization of visual features.

| Dataset | VLM Zero-Shot | | Fine-Tuning (FT) | | Linear Probe (LP) | |
| --- | --- | --- | --- | --- | --- | --- |
| | ACC | F1 | ACC | F1 | ACC | F1 |
| Derm7pt | 12.77 | 15.05 | 44.81 | 22.98 | **50.16** | **26.59** |
| eSkinHealth | 13.76 | 10.33 | 57.74 | 39.24 | **62.77** | **42.90** |
| Fitzpatrick17k | 10.05 | 8.27 | 42.84 | 30.49 | **58.88** | **46.22** |

Table 10: Effectiveness of the "Top-5 to Top-1" reranking strategy. This table compares three methods: the baseline zero-shot VLM, a standard k-shot linear probe (LP Top-1), and our two-stage approach. In our method, a k-shot linear probe selects the Top-5 candidates, which are then used by the VLM to produce a final prediction. The results show that our two-stage strategy consistently outperforms the direct linear probe's Top-1 accuracy in the low-data regime.

| Dataset | 0-shot Top-1 | 1-shot | | | 2-shot | | | 4-shot | | | 8-shot | | | 16-shot | | |
|---------|------|------|------|--------|------|------|--------|------|------|--------|------|------|--------|------|------|--------|
| | | Top1 | Top5 | 0-Shot | Top1 | Top5 | 0-Shot | Top1 | Top5 | 0-Shot | Top1 | Top5 | 0-Shot | Top1 | Top5 | 0-Shot |
| Derm7pt | 16.46 | 12.96 | 52.25 | 20.02 | 18.53 | 58.03 | 25.73 | 21.06 | 61.42 | 32.80 | 28.66 | 72.35 | 35.56 | 38.53 | 82.68 | 37.95 |
| eSkinHealth | 13.35 | 23.51 | 55.75 | 30.98 | 27.99 | 63.84 | 33.67 | 35.04 | 69.70 | 41.22 | 38.80 | 73.96 | 42.05 | 43.04 | 78.41 | 43.17 |
| Fitzpatrick17k | 16.73 | 17.63 | 50.58 | 25.35 | 23.08 | 59.50 | 27.42 | 31.01 | 67.67 | 33.68 | 38.43 | 75.88 | 37.82 | 45.05 | 80.99 | 38.01 |
| SD-260 | 10.81 | 11.48 | 26.89 | 22.31 | 17.90 | 38.61 | 25.02 | 26.97 | 51.92 | 33.50 | 37.12 | 63.61 | 42.33 | 49.01 | 76.72 | 44.72 |
| PAD-UFES-20 | 46.91 | 34.85 | 93.99 | 58.33 | 38.56 | 96.70 | 67.60 | 45.36 | 98.33 | 77.85 | 53.33 | 98.63 | 80.31 | 59.90 | 98.99 | 91.11 |

Table 11: Skin Condition Distribution for Fitzpatrick17k

| Skin Condition | Real Training | Real Test | Synthetic |
|----------------|---------------|-----------|-----------|
| Acne (ACN) | 92 | 91 | 93 |
| Actinic Keratosis (AK) | 88 | 87 | 164 |
| Allergic Contact Dermatitis (ACD) | 215 | 215 | 181 |
| Basal Cell Carcinoma (BCC) | 234 | 234 | 154 |
| Eczema (ECZ) | 102 | 102 | 166 |
| Erythema Multiforme (EM) | 118 | 118 | 155 |
| Folliculitis (FOL) | 171 | 171 | 114 |
| Granuloma Annulare (GA) | 106 | 105 | 148 |
| Keloid (KEL) | 78 | 78 | 135 |
| Lichen Planus (LP) | 246 | 245 | 151 |
| Lupus Erythematosus (LE) | 205 | 205 | 172 |
| Melanoma (MEL) | 130 | 131 | 155 |
| Mycosis Fungoides (MF) | 91 | 91 | 165 |
| Pityriasis Rosea (PR) | 96 | 97 | 156 |
| Prurigo Nodularis (PN) | 85 | 85 | 152 |
| Psoriasis (PSO) | 326 | 327 | 165 |
| Sarcoidosis (SAR) | 174 | 175 | 162 |
| Scabies (SCA) | 170 | 169 | 176 |
| Squamous Cell Carcinoma (SCC) | 290 | 291 | 175 |
| Vitiligo (VIT) | 83 | 83 | 161 |
| Total | 3100 | 3100 | 3100 |

Table 12: **Dataset details.**

| Dataset | # Classes | Train Size | Test Size | Class Names |
|---|---|---|---|---|
| Derm7pt | 14 | 413 | 395 | clark nevus, melanoma, reed or spitz nevus, seborrheic keratosis, basal cell carcinoma, vascular lesion, lentigo, blue nevus, dermal nevus, dermatofibroma, combined nevus, congenital nevus, miscellaneous, recurrent nevus |
| eSkinHealth | 24 | 2,714 | 2,676 | buruli ulcer, scabies, yaws, prurigo nodularis, tinea corporis, erysipelas, tinea capitis, leprosy, impetigo, necrotizing fasciitis, contact dermatitis, lichen planus, tinea versicolor, folliculitis, chickenpox, acne, vitiligo, abscess, keratosis, herpes zoster, atopic dermatitis, eczema, lipome, mycetoma |
| Fitzpatrick17k | 20 | 3,100 | 3,100 | psoriasis, squamous cell carcinoma, lichen planus, basal cell carcinoma, allergic contact dermatitis, lupus erythematosus, sarcoidosis, folliculitis, scabies, melanoma, erythema multiforme, granuloma annulare, eczema, pityriasis rosea, mycosis fungoides, acne, actinic keratosis, prurigo nodularis, vitiligo, keloid |
| SD-260 | 260 | 10,362 | 10,238 | abrasion, acne excoriee, acne keloidalis nuchae, acne vulgaris, acrokeratosis verruciformis, actinic solar damage, actinex treatment, actinic cheilitis, actinic keratosis, cutis rhomboidalis nuchae, actinic favre-racouchot, pigmentation, actinic solar elastosis, solar purpura, telangiectasia, actinic wrinkles, acute eczema, allergic contact dermatitis, alopecia areata, anagen effluvium, androgenetic alopecia, angiofibroma, angiokeratoma, angioma, angular cheilitis, aphthous ulcer, apocrine hydrocystoma, arsenical keratosis, atopic dermatitis, balanitis xerotica obliterans, basal cell carcinoma, "beaus lines", "beckers nevus", "behcets syndrome", benign keratosis, blue nevus, bowenoid papulosis, "bowens disease", cafe au lait macule, callus, candidiasis, cellulitis, chalazion, cherry angioma, clubbing of fingers, combined nevus, compound nevus, congenital nevus, contact dermatitis, "crowes sign", cutanea larva migrans, cutaneous horn, cutaneous leishmaniasis, cutaneous t-cell lymphoma, cutis marmorata, darier-white disease, dermatofibroma, dermatomyositis, dermatosis papulosa nigra, desquamation, digital fibroma, dilated pore of winer, discoid lupus erythematosus, disseminated actinic porokeratosis, drug eruption, dry skin eczema, dyshidrosiform eczema, dysplastic nevus, eccrine poroma, eczema, epidermal nevus, epidermoid cyst, epithelioma adenoides cysticum, erythema ab igne, erythema annulare centrifugum, erythema craquele, erythema multiforme, exfoliative erythroderma, factitial dermatitis, favre-racouchot, fibroma, fibroma molle, fixed drug eruption, follicular mucinosis, follicular retention cyst, fordyce spots, frictional lichenoid dermatitis, ganglion, geographic tongue, granulation tissue, granuloma annulare, green nail, guttate psoriasis, hailey-hailey disease, half and half nail, halo nevus, hand foot mouth disease, herpes gestationis, herpes simplex virus, herpes zoster, hidradenitis suppurativa, hirsutism, histiocytosis x, hyperkeratosis palmaris et plantaris, hypertrichosis, ichthyosis, ichthyosis vulgaris, id reaction, impetigo, infantile atopic dermatitis, insect bite, |

| Dataset | # Classes | Train Size | Test Size | Class Names |
|---------|-----------|------------|-----------|-------------|
|  |  |  |  | intradermal nevus,inverse psoriasis, ischemia, junction nevus, keloid, keratoacanthoma, keratolysis exfoliativa of wende, keratosis pilaris, kerion, koilonychia, "kyrles disease", leiomyoma, lentigo maligna melanoma, lentigo simplex, leprosy, leukemia cutis, leukocytoclastic vasculitis, leukonychia, lichen planus, lichen sclerosis et atrophicus, lichen simplex chronicus, lichen spinulosis, linear epidermal nevus, lipoma, livedo reticularis, lymphangioma circumscriptum, lymphocytic infiltrate of jessner, lymphocytoma cutis, lymphomatoid papulosis, mal perforans, malignant melanoma, median nail dystrophy, melasma, metastatic carcinoma, milia, molluscum contagiosum, morphea, mucha-habermann disease, mucous membrane psoriasis, myxoid cyst, nail cosmesis, nail dystrophy, nail nevus, nail psoriasis, nail ridging, nail trauma, neurodermatitis, neurofibroma, neurotic excoriations, nevus cell nevus, nevus comedonicus, nevus incipiens, nevus sebaceous of jadassohn, nevus spilus, nummular eczema, onychogryphosis, onycholysis, onychomycosis, onychoschizia, paronychia, pearly penile papules, pediculosis pubis, pemphigus foliaceus, perioral dermatitis, photodermatitis, pilomatrixoma, pincer nail syndrome, pitted keratolysis, pityriasis alba, pityriasis rosea, pityriasis rubra pilaris, pityriasis versicolor, pityrosporum folliculitis, poikiloderma atrophicans vasculare, pomade acne, porokeratosis of mibelli, port wine stain, pseudofolliculitis barbae, pseudorhinophyma, psoriasis, pterygium inversum unguis, pustular psoriasis, pyoderma gangrenosum, pyogenic granuloma, racquet nail, radiodermatitis, rhinophyma, rosacea, scabies, scalp psoriasis, scar, scarring alopecia, "schambergs disease", sebaceous gland hyperplasia, seborrheic dermatitis, seborrheic keratosis, skin tag, solar elastosis, solar lentigo, spindle cell nevus, squamous cell carcinoma (scc), stasis dermatitis, stasis edema, stasis ulcer, steroid acne, steroid atrophy, steroid striae, steroid use, stomatitis, strawberry hemangioma, striae, subacute cutaneous lupus erythematosus, subungual hematoma, superficial actinic porokeratosis, syringoma, systemic lupus erythematosus, "terrys nails", thermal burn, tick bite, tinea capitis, tinea corporis, tinea cruris, tinea faciale, tinea incognito, tinea manus, tinea pedis, tinea versicolor, toe deformity, traction alopecia, trichilemmal cyst, trichoepithelioma, trichofolliculoma, trichostasis spinulosa, trichotillomania, tuberous sclerosis, twenty nail dystrophy, ulcer, urticaria, uvl burn, varicella, verruca vulgaris, viral exanthem, virilization, vitiligo, "von recklinghausens disease", wart, wound infection, xerosis, x-linked ichthyosis |
| PAD-UFES-20 | 6 | 1,134 | 1,164 | actinic keratosis, basal cell carcinoma, melanoma, nevus, seborrheic keratosis, squamous cell carcinoma |

Table 13: **Skin condition descriptions used in this study.**

| Condition | Description |
| --- | --- |
| acne | Face/chest/back comedones and inflamed papules/pustules ±nodules; red or skin-colored with black/white heads; oily, ±crust. |
| actinic keratosis | Sun-exposed rough, scaly flat/slightly raised papule ¡1 cm; pink/red/brown; gritty sandpaper feel. |
| allergic contact dermatitis | At contact sites, ill-defined pink-red (darker on dark skin) patches/plaques ±vesicles/edema; weepy/crusty/scaly surface. |
| basal cell carcinoma | Sun-exposed pearly/waxy papule or thin scaly patch with rolled edge ±central ulcer; translucent or brown/black; smooth/shiny ±crust. |
| eczema | Flexural dry itchy ill-defined patches/plaques ±tiny vesicles; red/pink or purple/gray on dark skin; flaky ±lichenified. |
| erythema multiforme | Acral target lesions—round 1–3 cm with dark center, pale ring, red outer rim—mostly flat ±blister. |
| folliculitis | Hair-bearing sites with clustered 2–5 mm follicle-centered pustules/red papules; red/darker base with white/yellow pus; dome-shaped ±crust. |
| granuloma annulare | Hands/feet/wrists/ankles smooth firm non-scaly papules forming annular rings; skin-colored/pink/red (purple on dark skin). |
| keloid | Over scars on chest/shoulders/earlobes etc., shiny firm hairless raised irregular growth extending beyond wound; pink/red or darker. |
| lichen planus | Wrists/ankles etc. flat-topped polygonal 2–10 mm violaceous papules/plaques with fine white Wickham striae; shiny ±scale. |
| lupus erythematosus | Malar 'butterfly' smooth pink rash ±discoid coin-shaped scaly scarred plaques on scalp/ears; red or hyperpigmented; rough if discoid. |
| melanoma | Anywhere (palms/soles/nails in dark skin) asymmetric lesion with irregular borders, color variegation, often ¿6 mm; becomes raised/crusted/ulcerated. |
| mycosis fungoides | Non-sun areas with dry scaly patches → thicker scaly plaques (±smooth tumor nodules); pink-red to brown/darker; irregular. |
| pityriasis rosea | Trunk herald patch then multiple smaller ovals along skin lines; pink/salmon (gray/brown/purple on dark skin) with fine collarette scale. |
| prurigo nodularis | Reachable areas with multiple very itchy 1–3 cm firm nodules, often crusted/scabbed on top; pink/red/brown/black; thick/rough with excoriations. |
| psoriasis | Elbows/knees/scalp/lower back well-demarcated plaques with thick silvery/gray scale; pink/red or purple/dark brown; dry/flaky (Auspitz sign). |
| sarcoidosis | Face/shins/scars with smooth firm plaques/nodules/patches; purplish/red-brown or lighter/darker areas; rubbery; shin nodules are tender. |
| scabies | Finger webs/wrists/waist/genitals etc. with 5–15 mm wavy burrows plus clustered 1–2 mm itchy papules/vesicles; excoriated/crusted. |
| squamous cell carcinoma | Sun-exposed or scarred sites with firm scaly/crusted nodule/plaque/ulcer ±raised border/central depression; pink/red or darker; rough, may bleed. |
| vitiligo | Sharply bordered depigmented patches (often symmetric) on face/hands/feet/genitals; chalk-white contrast; normal texture without scale. |
| nevus | Anywhere well-circumscribed round/oval macule/papule/nodule with smooth borders; uniform tan/brown/black or skin-colored; smooth flat or dome-raised. |
| seborrheic keratosis | Anywhere except palms/soles 'stuck-on' waxy papule/plaque with sharp borders; tan to dark brown/black or mixed; warty/greasy with keratin plugs. |
| buruli ulcer | Limb lesion evolves from painless nodule/plaque/edema to undermined necrotic ulcer; circular/irregular with yellow-white base and violaceous/brown edge; moist ±slough. |
| yaws | Leg/foot raspberry-like papilloma ±ulcer then multiple papules/plaques/shallow ulcers; dome-lobulated; bright red → red-brown/yellow; verrucous/granular ±crust. |
| tinea capitis | Scalp alopecic scaly ring-like patch with broken hairs/black dots ±boggy kerion or yellow scutula; gray-white/red-brown; fine scale or boggy crust. |
| tinea corporis | Exposed skin annular plaques with raised scaly active border and central clearing; polycyclic/serpiginous; rim scaly, center smoother. |
| erysipelas | Lower legs/face sharply demarcated lobulated cellulitic plaque ±tense bullae; fiery red/violaceous; smooth tense shiny peau d'orange. |
| leprosy | Cool sites hypopigmented/coppery patches with sensory loss or plaques/nodules/diffuse infiltration; round/oval with raised rim; dry hairless/anhidrotic or thick shiny. |
| necrotizing fasciitis | Rapidly enlarging ill-defined limb/trunk plaque with dusky patches, ecchymoses and flaccid bullae → black necrotic eschar; tense shiny then leathery. |
| impetigo | Perioral/exposed erosions with honey-colored crust after vesicles/pustules (bullous form flaccid blisters); round/oval, coalescent; moist → sticky crust. |

Table 14: Basal Cell Carcinoma (BCC): visual checklist

| Section | Details |
|---------|---------|
| **Location** | • Sun-exposed areas most commonly:
• Face (nose, cheeks, forehead), ears, neck
• Scalp (esp. with thinning hair), upper chest/shoulders, back, hands, arms |
| **Lesion type (morphology)** | • "Pearly"/waxy bump (common presentation)
• Papillary: raised, dome-shaped bump
• Nodular: firm, raised nodule
• Superficial infiltrating: flat, scaly or crusted patch/plaque
• Morpheaform: indurated, scar-like plaque
• Micropapillary: flat/scaly with small raised papules
• Infiltrating BCC: flat, scaly lesion that can mimic other dermatoses |
| **Shape** | • Often irregular with poorly defined borders
• Typically asymmetrical
• Evolving in size, shape, or color over time |
| **Color** | • Pink or red; may be white/waxy
• Brown/tan or (less commonly) black
• Blue/purple hues if ulcerated |
| **Texture** | • Smooth or waxy; may be scaly or crusted
• Ulcerated surface can occur and is concerning |

| Condition | Description |
|-----------|-------------|
| tinea versicolor | Upper trunk/shoulders/neck flat hypo- or hyperpigmented macules/patches with fine powdery 'bran' scale accentuated by scraping; coalescing map-like. |
| varicella | Face/trunk then scalp/limbs crops of itchy 2–4 mm thin-walled vesicles on pink/dark base → crusted scabs; discrete ±coalescent; glistening then flaky. |
| abscess | Axillae/buttocks/groin etc. tender dome-shaped red-violaceous nodule that becomes fluctuant with pointing yellow/white center; shiny tense skin; purulent drainage after rupture. |
| atopic dermatitis | Flexures/hands/eyelids etc. itchy ill-defined papules→plaques with acute weeping/crust or chronic lichenification; pink-red or purple-brown/gray; dry rough. |
| keratosis pilaris | Outer arms/thighs/cheeks tiny follicular keratin plugs ('goose-bump' papules) in indistinct patches; skin-colored to red; rough sandpapery dry feel. |
| herpes zoster | Single-dermatome band of clustered clear vesicles on pink/purple base not crossing midline → pustules/crusts; 2–4 mm; tense/glossy then adherent crust. |
| lipoma | Trunk/neck/limbs soft smooth mobile subcutaneous dome/oval nodule with normal overlying color; rubbery/doughy consistency. |
| mycetoma | Foot/leg/hand firm lobulated subcutaneous mass with multiple draining sinus openings extruding colored grains; overlying skin normal→hyperpigmented; crusted seepage. |
| lymphatic filariasis | Lower legs/feet (±arms/genitals) chronic lymphedema to elephantiasis with column-like limb, bulbous foot and verrucous corrugated plaques; hyper/hypopigmented; thick 'mossy' hyperkeratosis. |

Table 15: Lupus Erythematosus (LE): visual checklist

| Section | Details |
| --- | --- |
| **Location** | • **Sun-exposed areas** most common: face (cheeks, nose, forehead), ears, neck, chest, upper arms
• **Mucous membranes**: oral cavity (inner cheeks, gums, tongue), nose, eyelids
• **Other sites** (less common): scalp, nails, genital area |
| **Lesion type** | • **Malar (butterfly) rash**: flat/slightly raised erythema across cheeks and nasal bridge
• **Discoid lupus**: chronic coin-shaped, raised, scaly, scarring plaques; may leave hypo-/hyperpigmentation
• **Photosensitivity**: rash/exacerbation after sun exposure
• **Oral ulcers**: shallow ulcers on inner cheeks, gums, or tongue
• **Livedo reticularis**: net-like reddish-blue mottling (legs/arms/face)
• **Vasculitis**: purpura, petechiae, ulcers
• **Alopecia**: diffuse thinning or patchy hair loss
• **Raynaud's phenomenon**: digits turn white/blue with cold or stress |
| **Shape** | • Malar rash: butterfly-shaped, may have central clearing
• Discoid: coin-shaped, raised, scaly plaques
• Oral ulcers: round/oval
• Livedo: reticular (net-like) pattern
• Purpura/petechiae: small pinpoint to macular spots
• Alopecia: patchy or generalized |
| **Color** | • Malar: red, sometimes with central clearing
• Discoid: red/erythematous; may heal with hypo- or hyperpigmentation
• Oral ulcers: red to violaceous
• Livedo: reddish-blue mottling
• Purpura/petechiae: red or purple
• Alopecia: variable (depends on cause and background skin) |
| **Texture** | • Malar: flat or slightly raised
• Discoid: raised, scaly; may become crusted; can scar
• Oral ulcers: smooth base, slightly irregular margins
• Livedo: smooth surface with reticular pattern
• Purpura/petechiae: flat to slightly raised
• Alopecia: variable; scalp may be smooth or scaly depending on subtype |

