# OpenReview forum: "Are Medical Vision–Language Foundation Models Ready for Dermatology"
_ICLR.cc/2026/Conference — Submitted to ICLR 2026_

### Official Review · Reviewer_bjJR · 2025-10-31

**Soundness:** 2
**Presentation:** 2
**Contribution:** 2
**Rating:** 2
**Confidence:** 4

**Summary:**

This paper investigates the performance gap between the vision encoder capabilities and end-to-end diagnostic accuracy of MedGemma-4B in dermatology.

**Strengths:**

1. The performance gap between vision encoders and full VLMs in safety-critical medical applications is a significant issue that deserves investigation.
2. Evaluation across five diverse datasets (SD-260, Fitzpatrick17k, Derm7pt, eSkinHealth, PAD-UFES-20) covering different diseases, demographics, and data characteristics.

**Weaknesses:**

1. The generalizability claims are unsupported without evaluation on other VLMs (GPT-4V, LLaVA-Med, SkinGPT-4, SkinVL, Med-Flamingo, etc.).
2. Missing comparison with standard prompting baselines (few-shot prompting, chain-of-thought without describe-first).
3. The paper should clearly distinguish between truly training-free strategies (in-context, describe-then-decide) and the hybrid approach. There is misleading "Training-Free" and "Zero-Shot" Framing.
4. No confidence intervals, standard deviations, or significance tests reported despite "averaged over five runs."
5. No analysis of which improvements are statistically significant. And no ablation on key hyperparameters (why 8-shot?)
6. Table 1 shows LoRA fine-tuning dramatically underperforms linear probes (39.50% vs 57.22% on Derm7pt) . It is a surprising result deserves deep investigation but receives minimal analysis.
7. Evidence is circumstantial. The model describing diseases it cannot diagnose doesn't definitively prove vision training data issues - could be alignment/integration problems.
8. No evaluation of whether generated descriptions are actually useful for clinical decision-making. Only 100 images evaluated by a single dermatologist (no inter-rater reliability).

**Questions:**

1. Can you evaluate your strategies on at least 2-3 other medical VLMs (e.g., GPT-4V, LLaVA-Med, one other dermatology-specific model)?
2. What percentage of test cases have the correct diagnosis outside the Top-5 candidates? How does this affect overall potential accuracy?
3. Can you provide statistical significance tests (e.g., paired t-tests or McNemar's test) for the improvements in Table 4?
4. Why does LoRA fine-tuning perform so much worse than a simple linear probe? Have you tried different LoRA ranks, learning rates, or full fine-tuning?
5. Can you provide concrete examples of "reasoning capabilities" that are lost during fine-tuning? Include qualitative comparisons of model outputs.

---

> ### Author Response · Authors · 2025-11-25
> **Response to Reviewer bjJR [1/3]**
>
> We thank the reviewer for recognizing the significance of the research problem we are investigating in this work and the comprehensiveness of our evaluation. We also sincerely appreciate the constructive feedback on our submission.
>
> ## W1: Generalizability of the Claims
>
> Thank you for this suggestion. We agree that establishing the universality of the observed performance gap is vital. To address this, we examined the additional models suggested by the reviewer. We prioritized models that are open-source and dermatologically pre-trained and extended our evaluation to SkinVL (2025). We selected SkinVL as it represents a distinct architectural lineage (LLaVA-based vs. MedGemma’s Gemma-based) and is a current state-of-the-art model in this domain. (Note that Skin-GPT4 didn't release it's full model due to privacy constraint.)
>
> | **Models**       | **Open-sourced** | **Dermatologically Pre-trained** | **Language Base** | **Year** |
> | ---------------- | ---------------- | -------------------------------- | ----------------- | -------- |
> | **GPT-4V**       | N                | N                                | GPT-4             | 2023     |
> | **LLaVA-Med**    | Y                | N                                | LLaVA             | 2023     |
> | **Med-Flamingo** | Y                | N                                | LLaMA-7B          | 2023     |
> | **SkinGPT-4**    | Y                | Y                                | LLaMa-2-13B       | 2024     |
> | **SkinVL**       | Y                | Y                                | LLaVA-Medv1-7B    | 2025     |
>
> Consistent with our findings on MedGemma, we observed that SkinVL also exhibits a significant performance gap. As shown below, a Linear Probe (LP) on the vision encoder significantly outperforms the VLM's zero-shot capabilities and, notably, even outperforms full fine-tuning (FT) in several metrics. This confirms that **under-utilization of the vision encoder is a pervasive issue**, not a quirk of MedGemma.
>
> | **Data Details**  | **0-Shot ACC** | **0-Shot F1** | **FT ACC** | **FT F1** | **LP ACC** | LP F1 |
> | ----------------- | -------------- | ------------- | ---------- | --------- | ---------- | ----- |
> | **Derm7pt**       | 12.77          | 15.05         | 44.81      | 22.98     | 50.16      | 26.59 |
> | **eSkinHealth**   | 13.76          | 10.33         | 57.74      | 39.24     | 62.77      | 42.90 |
> | **Fitzpatrik17k** | 10.05          | 8.27          | 42.84      | 30.49     | 58.88      | 46.22 |
>
> We further tested our proposed inference strategies on SkinVL. Due to SkinVL’s context window constraints, we adapted the pipeline to perform "Top-5 to Top-1" filtering first, followed by "In-Context" and "Describe-then-Decide" prompting. It is important to note that we adapted the pipeline's execution order to strictly adhere to SkinVL's input token constraints.
>
> |                           | **Derm7pt** | **eSkinHealth** | **Fitzpatrick17k** |
> | ------------------------- | ----------- | --------------- | ------------------ |
> | **direct_answer**         | 12.77       | 23.76           | 10.05              |
> | **+top_5 to top_1**       | 29.66       | 40.25           | 58.37              |
> | **+in-context**           | 31.19       | 44.03           | 62.97              |
> | **+describe-then-decide** | 32.56       | 44.89           | 63.91              |
>
> The high baseline for Fitzpatrick17k in the Top-5 method is likely due to the dataset's inclusion in SkinVL's pre-training data; however, the relative improvement trend remains consistent with our MedGemma findings.
>
> These new experiments confirm that (1) The performance gap between vision encoders and VLMs is an universal phenomenon for dermatological VLM and (2) Our proposed training-free pipeline successfully generalizes to LLaVA-based architectures, yielding consistent performance gains. We will incorporate these new results into the Appendix to robustly address the question of generalization.
>
> ## W2: Missing Comparison with Standard Prompting Baselines
>
> We appreciate this feedback! While we acknowledge the value of such comparisons, we respectfully clarify that the primary contribution of this work is not to propose a general-purpose prompting strategy that simply outperforms standard baselines. Rather, our core value lies in systematically diagnosing the root causes of the performance gap (Hypotheses 1–3) and providing targeted, hypothesis-driven interventions to address them. Our strategies were designed specifically to isolate and resolve these identified failure modes rather than to compete on prompt techniques. Nevertheless, we agree that benchmarking against standard strategies is crucial for the broader deployment of medical VLMs, and we will highlight this for future investigation in the Discussion Section.

---

> ### Author Response · Authors · 2025-11-25
> **Response to Reviewer bjJR [2/3]**
>
> ## W3: Reframing "Training-free"
> Thank you for this crucial clarification! We agree that our initial terminology was imprecise. We will re-frame the pipeline using the more precise term **"Fine-Tuning-Free"** rather than "Training-Free." This distinction highlights the core advantage of our approach: unlike LoRA or full fine-tuning—which updates the VLM's weights and compromises its reasoning capabilities—our "fine-tuning-free," inference-time method keeps the VLM **completely frozen**. By training only a lightweight, external linear probe, we combine the discriminative power of the vision encoder with the preserved, high-quality reasoning priors of the foundation model. This is especially practical for medical practitioners with limited computing resources and training data. We appreciate the reviewer pointing this out, as this clearer framing strengthens our core argument.
>
> ## W4, W5, Q3: Statistical Variance & Ablation Study on # Shots
> We thank the reviewer for emphasizing the need for statistical rigor. To address this, we will update the manuscript to include standard deviations across all tables. Regarding the ablation study on the number of shots: We selected the 8-shot setting based on the analysis presented in Table 2, where 8-shot maximizes the utility of our pipeline. At 8-shot, the vision encoder is strong enough to capture the correct diagnosis in the Top-5 (high recall), but not yet perfect at Top-1 (low precision), creating the ideal scenario for the VLM to apply its reasoning for re-ranking. Beyond 8 shots, the vision encoder becomes self-sufficient, and the added value of VLM reasoning diminishes.
>
> ## W6, Q4: Performance Gap Between LoRA FT & LP
> These questions get to the core of our paper's motivation. Our hypothesis is objective mismatch.
> - The Linear Probe (LP) is a simple classifier on the frozen, high-quality MedSigLIP vision features. It excels at its one simple task.
> - The LoRA fine-tuning attempts to update the weights of the entire generative VLM to perform this simple classification task. This likely causes two problems: (1) Catastrophic forgetting (as cited, Zhai et al., 2023b) of the model's rich generative reasoning priors, and (2) Objective misalignment, as the model is torn between its original generative purpose and its new, narrow classification goal.
>
> We will make the connection explicit in the manuscript and include the ablation study on different ranks.
>
> ## W7: Circumstantial Evidence for Hypothesis 1
> We thank the reviewer for this insightful critique. We agree that without access to MedGemma’s private pre-training data, the evidence for "Distribution Mismatch" (H1) in our initial submission was deductive rather than direct. However, we argue that this deduction is robust based on triangulation, and we have now bolstered this with direct evidence from the open-source SkinVL model.
>
> 1. Triangulating the Cause in MedGemma (Deductive Evidence): We view "alignment/integration failure" as the *mechanism* of the error, but "data imbalance" as the *root cause* driving that failure. Our analysis relied on three converging signals:
>    - Prediction Bias (Fig. 2a): The model exhibits a massive skew toward common classes (e.g., Basal Cell Carcinoma) even when visual features are distinct, suggesting a learned prior probability.
>    - Intact Textual Knowledge (Sec 3.1): When prompted, the model can accurately describe rare diseases it fails to diagnose. This isolates the failure: it is not a lack of conceptual knowledge (Language branch), but a failure to trigger that concept from visual input (Vision-Language mapping).
>    - Efficacy of Intervention (Fig. 2b): Manually bridging the visual-to-label gap via "in-context" descriptions corrects the error. If the issue were a fundamental architectural breakage (integration failure), providing text descriptions would likely not suffice to recover the correct diagnosis.
>
> 2. To address your concern that this might be a model-specific integration issue rather than a data distribution issue, we tested this hypothesis on SkinVL, whose training data is public (consisting of clinical images from SCIN and Fitzpatrick17k). If H1 is correct, the model’s inference bias should directly mirror its training data distribution.
>    - Training Distribution: In the SkinVL training corpus, Lichen Planus and Pityriasis Rosea are among the heavily represented classes.
>    - Inference Bias: When we tasked SkinVL with free-form QA on the Fitzpatrick17k test set, it displayed a heavy bias toward these exact classes, predicting Lichen Planus (20.22% of all predictions) and Pityriasis Rosea (15.65%) at rates far exceeding their actual prevalence in the test set.
>
> This direct correlation between training data frequency and inference prediction bias in SkinVL provides strong empirical support for our hypothesis: the "alignment" breaks specifically where the visual training data is sparse, leading the model to fall back on the most visually familiar classes.

---

> ### Author Response · Authors · 2025-11-25
> **Response to Reviewer bjJR [3/3]**
>
> ## W8: Generated Descriptions for Clinical Decision-making
>
> We agree that assessing clinical utility is the ultimate goal for any medical AI system. However, we respectfully clarify that the primary scope and contribution of this work is to systematically diagnose the root causes of the performance gap in medical VLMs (e.g., modality misalignment, language prior dominance) and to propose targeted inference-time solutions. The human evaluation in Figure 6 was intended as a proof-of-concept to validate Hypothesis 3: demonstrating that the model is capable of accurate description (scoring high on structured criteria) even when it fails at zero-shot diagnosis. While we recognize that a full clinical utility study involving multiple experts and inter-rater reliability analysis is critical for real-world deployment, such an extensive study is beyond the scope of this technical analysis. We will update Appendix to explicitly state this limitation and to highlight that a comprehensive clinical decision-making evaluation is a necessary and immediate priority for future work.
>
>
> ## Q2: Percentage of Test Cases Outside the Top-5 Candidates
>
> The "Top-5 (LP few-shot)" line in Figure 1 and the "LP Top-5" columns in Table 2 provide this exact number. For example, in Table 2 for our 8-shot setting:
>
> - On Derm7pt, the LP Top-5 accuracy is 72.35%.
> - On PAD-UFES-20, it is 98.63%.
>
> This means that for Derm7pt, the correct diagnosis is outside the Top-5 candidates in 27.65% of cases. This Top-5 accuracy (72.35%) represents the theoretical performance ceiling for our "Top-5 to Top-1" strategy. Our method's goal is to be the best possible re-ranker within that set. Could you elaborate on your question if we misunderstood it?
>
> ## Q5: Concrete Examples of Lost "Reasoning Capabilities"
>
> We thank the reviewer for highlighting this. We will add a qualitative comparison in Appendix to illustrate this phenomenon. While LoRA fine-tuning can improve classification accuracy compared to the zero-shot baseline, it often causes the model to suffer from catastrophic forgetting of its generative capabilities. The fine-tuned model tends to "collapse" into a simple classifier, outputting short, label-focused answers while losing the ability to provide detailed visual descriptions or nuanced reasoning. Beyond preserving these reasoning capabilities, our choice of a "fine-tuning-free" solution is also driven by practical constraints. In many real-world medical settings, practitioners lack the annotated data for instruction tuning or the computational resources required to fine-tune a VLM effectively.

---

### Official Review · Reviewer_6Nrz · 2025-10-31

**Soundness:** 3
**Presentation:** 3
**Contribution:** 2
**Rating:** 4
**Confidence:** 4

**Summary:**

The paper investigates whether current medical vision-language foundation models (VLMs) are ready for dermatology applications. The authors identify three potential sources of failure: 1. distribution mismatch between training and target data, 2. overreliance on language priors, and 3. misalignment between vision and language objectives.
To address these, they propose two prompt-based inference strategies—adding clinical context descriptions and a “Describe-then-Decide” reasoning approach—and a two-stage pipeline that combines a visual probe with VLM-based reasoning. The motivation is well-grounded: existing medical VLMs often perform poorly in diagnosis because they rely too heavily on textual correlations instead of actual image evidence.

**Strengths:**

1. The paper tackles a highly relevant problem: understanding and mitigating the role of language priors in medical VLMs. This is crucial for deploying such models in safety-critical clinical contexts.

2. The proposed techniques, though simple, lead to consistent accuracy gains across multiple dermatology datasets and improve the performance by a decent margin.

**Weaknesses:**

1. The work focuses mainly on adjusting prompts and modifying the inference pipeline rather than introducing any novel algorithmic or modeling framework. Although the paper correctly identifies the entanglement between image and language priors as a core issue, it does not explore how to decouple these modalities at the representation or optimization level.

2. The proposed strategies (prompt reformulation and two-stage decision) highly tailored to dermatology tasks and to specific models such as MedGemma. It is unclear whether these improvements would generalize to other modalities, datasets, or medical domains.

3. While the paper attributes VLM underperformance to language priors, the presented evidence does not conclusively prove that the three identified factors are the primary causes. Other factors, such as inadequate visual feature alignment or data imbalance, might explain the same phenomena.

**Questions:**

1. Do we have any further experiments on other medical image datasets except the dematology datasets.
2. Do you think radiology dataset also have the same problem?

---

> ### Author Response · Authors · 2025-11-25
> **Response to Reviewer 6Nrz [1/2]**
>
> We thank the reviewer for recognizing the significance of the research problem studied in the work and the effectiveness of our proposed pipeline. We also sincerely appreciate the constructive feedback on our submission.
>
> ## W1: Question about Contribution
>
> Our focuses on inference-time strategies rather than proposing a new model architecture was a deliberate choice, as our primary goal was to provide actionable guidelines for practitioners to deploy large VLMs more effectively. While new architectures are vital, they require immense resources for pre-training. Our inference-time approach is designed to be immediately applicable by practitioners in resource-constrained settings (limited GPUs or annotated data), allowing them to significantly improve the safety and accuracy of existing state-of-the-art models like MedGemma without costly fine-tuning or architectural changes. Our core contribution is the systematic analysis of the performance gap (H1, H2, H3) and a targeted, data-efficient pipeline to address these specific failures. We believe this analysis provides a valuable diagnostic framework that can inform future architectural work, while our solutions offer a practical path to more reliable deployment today.
>
> ## W2, Q1: Generalization to Other Modalities/Datasets/Medical Domains
>
> We appreciate the reviewer’s concern regarding generalizability. While our experiments are currently scoped to dermatology, we respectfully argue that our findings have broad implications for two key reasons:
>
> 1. **The Strategic Value of the Domain**: Dermatology serves as an ideal testbed for investigating Vision-Language alignment because it is an inherently visual discipline with high inter-class similarity and intra-class diversity, demanding fine-grained visual discrimination. Therefore, solving modality misalignment and hallucination in this high-stakes, visually dependent field offers a non-trivial "stress test" for VLM architectures that is relevant to other image-intensive fields like pathology and radiology.
>
> 2. **The Universality of the Underlying Principles**: We posit that the root causes we identified are not unique to dermatology but are fundamental issues in current Vision-Language research. The three hypotheses we validated are highly generalizable to other medical domains:
>
> - H1 (Distribution Mismatch): "Long-tail" class distributions are endemic to almost all medical data (e.g., rare pathologies in radiology). Our "In-Context" strategy is a generalizable method for injecting domain knowledge for these rare classes.
> - H2 (Under-reliance on Vision): Hallucination and language-prior dominance are well-documented universal VLM issues. Our "Describe-then-Decide" strategy is a domain-agnostic technique to force visual grounding.
> - H3 (Objective Misalignment): The tension between a discriminative vision encoder and a generative LLM exists in all VLMs. Our "Inference-time Pipeline" is a model-agnostic architectural solution.
>
> To empirically demonstrate that these issues (and our solutions) are not specific to the MedGemma architecture, we extended our evaluation to SkinVL (2025). We selected SkinVL as it represents a distinct architectural lineage (LLaVA-based vs. MedGemma’s Gemma-based) and is a current state-of-the-art model in this domain.
>
> | **Models**       | **Open-sourced** | **Dermatologically Pre-trained** | **Language Base** | **Year** |
> | ---------------- | ---------------- | -------------------------------- | ----------------- | -------- |
> | **GPT-4V**       | N                | N                                | GPT-4             | 2023     |
> | **LLaVA-Med**    | Y                | N                                | LLaVA             | 2023     |
> | **Med-Flamingo** | Y                | N                                | LLaMA-7B          | 2023     |
> | **SkinGPT-4**    | Y                | Y                                | LLaMa-2-13B       | 2024     |
> | **SkinVL**       | Y                | Y                                | LLaVA-Medv1-7B    | 2025     |

---

> ### Author Response · Authors · 2025-11-25
> **Response to Reviewer 6Nrz [2/2]**
>
> Consistent with our findings on MedGemma, we observed that SkinVL also exhibits a significant performance gap. As shown below, a Linear Probe (LP) on the vision encoder significantly outperforms the VLM's zero-shot capabilities and, notably, even outperforms full fine-tuning (FT) in several metrics. This confirms that **under-utilization of the vision encoder is a pervasive issue**, not a quirk of MedGemma.
>
> | **Data Details**  | **0-Shot ACC** | **0-Shot F1** | **FT ACC** | **FT F1** | **LP ACC** | LP F1 |
> | ----------------- | -------------- | ------------- | ---------- | --------- | ---------- | ----- |
> | **Derm7pt**       | 12.77          | 15.05         | 44.81      | 22.98     | 50.16      | 26.59 |
> | **eSkinHealth**   | 13.76          | 10.33         | 57.74      | 39.24     | 62.77      | 42.90 |
> | **Fitzpatrik17k** | 10.05          | 8.27          | 42.84      | 30.49     | 58.88      | 46.22 |
>
> We further tested our proposed inference strategies on SkinVL. Due to SkinVL’s context window constraints, we adapted the pipeline to perform "Top-5 to Top-1" filtering first, followed by "In-Context" and "Describe-then-Decide" prompting.
>
> |                           | **Derm7pt** | **eSkinHealth** | **Fitzpatrick17k** |
> | ------------------------- | ----------- | --------------- | ------------------ |
> | **direct_answer**         | 12.77       | 23.76           | 10.05              |
> | **+top_5 to top_1**       | 29.66       | 40.25           | 58.37              |
> | **+in-context**           | 31.19       | 44.03           | 62.97              |
> | **+describe-then-decide** | 32.56       | 44.89           | 63.91              |
>
> The high baseline for Fitzpatrick17k in the Top-5 method is likely due to the dataset's inclusion in SkinVL's pre-training data; however, the relative improvement trend remains consistent with our MedGemma findings. These new experiments confirm that (1) The performance gap between vision encoders and VLMs is an universal phenomenon for dermatological VLM and (2) Our proposed fine-tuning-free pipeline successfully generalizes to LLaVA-based architectures, yielding consistent performance gains. We will incorporate these new results into the Appendix to robustly address the question of generalization.
>
> ## W3: Inconclusive Factors
>
> We thank the reviewer for this insightful comment. We would respectfully argue that the "other factors" mentioned are precisely the phenomena our three hypotheses identify and provide evidence for.
>
> - "Data imbalance" is exactly what we investigate in Hypothesis 1: Train-Test Distribution Mismatch. Our evidence (Fig. 2a) shows the model over-predicts common classes (BCC), and our probe (Sec 3.1) confirms the model has textual knowledge of rare classes (like LE) but fails to visually identify them, pointing to an imbalanced visual training distribution.
> - "Inadequate visual feature alignment" and "overreliance on language priors" are the core issues we analyze in Hypothesis 2 (Under-reliance on Vision) and Hypothesis 3 (Objective Misalignment).
>   - For H2, our evidence is direct: the model hallucinates explanations (Fig. 3) and, crucially, fails the Abstention Test (Fig. 4), diagnosing masked images 88% of the time, conclusively proving it is not aligned with the visual features and is defaulting to language priors.
>   - For H3, our evidence is the massive performance gap itself (Fig. 1) between the (strong) vision-only probe and the (weak) end-to-end VLM.
>
> Thus, we believe our three hypotheses provide a well-evidenced and specific breakdown of the very problems the reviewer has identified.
>
> ## Q2: Radiology Dataset
>
> Yes, we strongly believe radiology datasets exhibit the same fundamental problems. In radiology, models often face the "normalcy bias" (H1), where they default to predicting "No Acute Findings" due to the prevalence of healthy scans in training data. Furthermore, radiology reports are complex texts; a VLM might generate a plausible-sounding report based on patient history while ignoring subtle visual evidence of a nodule (H2/H3). Therefore, we are confident that our framework would be highly effective in radiology.

---

> > ### Comment · Reviewer_hfSy · 2025-11-27
> > **Post-rebuttal comments**
> >
> > The authors have adequately answered my questions and clarified a few misunderstandings from my side. While I still believe that the algorithmic innovation of the work is rather modest – a concern also reflected in the reviews by other referees – I am happy to uphold my positive overall rating, as I feel that the work is of interest to the growing community of scientists interested in applying VLMs for medical tasks.

---

> > > ### Author Response · Authors · 2025-12-04
> > > **Follow-up Response to Reviewer hfSy**
> > >
> > > We sincerely thank you for upholding the positive rating! We appreciate your recognition that this work is of significant interest to the growing community of scientists applying VLMs in medicine. We agree that while our proposed solution is algorithmically simple (by design, to remain fine-tuning-free), we believe its primary contribution lies in
> > >
> > > ​	(i) systematic analysis uncovering why VLMs fail to unleash the full potential of their vision branch, and
> > >
> > > ​	(ii) providing a practical framework for recovering this performance in resource-limited clinical settings.
> > >
> > > Thank you again for your time and constructive feedback, which has helped us improve the clarity and quality of our manuscript.

---

> ### Comment · Reviewer_6Nrz · 2025-11-28
> **Comments to the Response**
>
> Thank you for your effort, here is my updated review:
>
> 1. Mechanism Analysis vs. Test-Time Utility: First, while I acknowledge the utility of the proposed work at test-time, I was hoping for a more profound investigation and explanation of the underlying model mechanisms. Specifically, insights regarding the interaction between visual and language tokens, inherent architectural flaws, or other structural implications are lacking. Although the paper proposes some mitigation strategies, I still perceive them as somewhat superficial. Furthermore, given that the scope is strictly limited to specific modalities, it is difficult to see how this work offers broad reference value to the wider Medical VLM research community.
>
> 2. Limitations of Description-Guided Attention in Radiology: Second, while the authors repeatedly emphasize the characteristics of the dermatology modality, it appears they are avoiding a critical issue: the core method relies on using descriptions to guide VLM attention. However, in similar works—and in my own experience—it is evident that language descriptions often yield minimal benefit in certain medical modalities. This is not a universal solution. The authors have failed to provide experiments on additional modalities to refute this. Language-based guidance tends to perform better in modalities resembling natural images (e.g., endoscopy, dermoscopy), but rarely provides substantial gains in radiology, particularly with modalities like Ultrasound. Previous studies suggest that retraining is often required to effectively leverage descriptive text in these contexts. This reinforces my view that a purely test-time approach cannot thoroughly resolve the visual-language representation gap identified by the authors themselves.
>
> 3. Clarification on "Inadequate Visual Feature Alignment": Regarding my comment on "inadequate visual feature alignment," I wish to clarify my intention. I am not referring solely to language priors influencing decision-making, as discussed in Hypothesis II in the paper. I am well aware of related research in this area; for instance, a paper at NAACL 2025[1] pointed out that VLMs tend to rely more heavily on language tokens during long generation sequences or difficult recognition tasks, leading to hallucinations. Therefore, I actually strongly agree with the second issue raised by the authors.
>
> However, regarding the proposed mitigation strategy, as mentioned above, I find it limited and lacking in novelty. There is already a significant amount of similar research in Chain-of-Thought (CoT) work for general VLMs using similar strategy. The authors essentially prompt the model to generate descriptions to focus attention on lesion areas. My primary concern, however, is the validity of these generated descriptions. When satisfying visual features are difficult to identify, or when dealing with irregular shapes, language descriptions often become vague. Addressing this ambiguity is the critical challenge for medical imaging. Unlike in the general domain, where visual features and language are tightly aligned (thanks to massive pre-training data), medical data is scarce. Consequently, medical concept visual features and language descriptions are often misaligned, particularly in radiological imagery.
>
> Conclusion In summary, while this study raises some thought-provoking points, the issues identified regarding VLMs are not particularly novel to those who follow this field closely. The findings and solutions presented here are unlikely to be seen as groundbreaking by VLM researchers. Although I recognize this work as a pioneer in this specific medical subdomain and acknowledge its high level of completion, I maintain that significant polishing is still required given the issues outlined above. I would strongly encourage the authors to expand their scope to additional modalities; demonstrating broader applicability—even if the solution is imperfect—would be more valuable than merely improving metrics on a specific task.
>
> Therefore, I will maintain my original score.
>
> [1] Lee, Kang-il et al. “VLind-Bench: Measuring Language Priors in Large Vision-Language Models.” North American Chapter of the Association for Computational Linguistics (2024).

---

> > ### Author Response · Authors · 2025-12-04
> > **Follow-up Response to Reviewer 6Nrz**
> >
> > We sincerely thank the reviewer for the detailed engagement and for acknowledging the utility of our work at test-time, as well as our role as a "pioneer in this specific medical subdomain." We appreciate the reviewer’s perspective on the broader challenges of medical VLMs. However, we would like to offer three final clarifications to contextualize our contributions for the Area Chair and future readers.
> >
> > 1. **On Mechanistic Depth vs. Practical Utility**: While we agree that token-level mechanistic interpretability is a fascinating future direction, we respectfully argue that it is a distinct research objective from ours. Our work focuses on **clinical translation**: identifying the existence of the performance gap in state-of-the-art medical foundation models and providing an immediate solution. In the context of resource-limited clinical settings, the fact that our inference pipeline does not require the heavy computational cost or expert-annotated data needed for re-training is a strategic advantage. It allows practitioners to unlock the latent potential of existing models immediately.
> >
> > 2. **On Generalization to Modalities Like Radiology/Ultrasound**: We agree with the reviewer’s expert observation that language-guided attention is harder in modalities like Ultrasound compared to natural-image-like modalities like dermatology. However, we respectfully argue that **dermatology is a sufficiently complex and high-stakes domain that needs dedicated study**. The fact that description-guided attention works in dermatology (as proven by our results) but might fail in ultrasound does not invalidate our findings. Rather, it highlights that different medical modalities require tailored approaches. Demanding that a dermatology-focused solution immediately solve the visual-language representation gap in ultrasound expands the scope beyond this work.
> >
> > 3. **On the Validity of Descriptions**: The reviewer expresses concern that "language descriptions often become vague." We would like to point the AC and reviewer back to our Human Evaluation (Figure 6). We explicitly tested this concern. A board-certified dermatologist rated 100 generated descriptions and found them to be highly accurate (scoring 4-5/5) in describing lesion morphology, color, and texture etc.,. This empirical evidence contradicts the theoretical concern that the descriptions are vague—at least for the domain of dermatology. **The model can describe the features accurately; the failure mode we solved is that the model simply wasn't using those features for the final classification/diagonosis**.
> >
> > Conclusion: We are glad to note our consensus on the fundamental problem of language prior over-reliance. We believe that a study which **(1)** quantifies the vision-language gap across five dermatological datasets, **(2)** validates these findings on state-of-the-art VLMs for dermatology (MedGemma and SkinVL), and **(3)** provides a practical, fine-tuning-free solution, offers value to the community. We thank you again for the stimulating discussion.

---

### Official Review · Reviewer_hfSy · 2025-11-01

**Soundness:** 3
**Presentation:** 3
**Contribution:** 2
**Rating:** 6
**Confidence:** 4

**Summary:**

The presented work explores the ability of the MedGemma-4B vision-language model to effectively diagnose dermatological diseases. Initially, the authors show that MedGemma performs substantially worse than its MedSigLIP vision encoder tuned with linear probing. They postulate that this alignment gap cannot be overcome using model fine-tuning via LoRA. In response, they propose three training-free strategies to improve MedGemma’s performance on five different dermatological datasets. The first strategy includes short summaries of the diseases’ visual features in the prompt to MedGemma. The second strategy prompts MedGemma to first describe the visible pathological features in the provided image before linking these to a diagnosis. Finally, the authors split report generation into a two-stage pipeline: initially, the pre-trained image encoder is used to identify the five most likely diagnoses before the VLM is asked to make the final classification. Each of these strategies is shown to individually improve MedGemma’s performance, while their combination achieves the highest overall performance increase.

**Strengths:**

- Evaluating the performance of foundational vision-language models in real-world clinical settings is of interest to both the machine learning and medical scientific community.

- The proposed training-free strategies are shown to substantially improve the performance of MedGemma.

- The authors conduct extensive benchmarking experiments involving five different dermatological datasets as well as further human rater and ablation studies.

- The manuscript is pleasant to read with its clear narrative-driven structure, many informative figures and tables, and high-quality writing.

**Weaknesses:**

- Prompting strategies for VLMs have been widely researched. For example, requesting structured reports has already been proposed (Delbrouck et al. “Automated Structured Radiology Report Generation.” Proceedings of the 63rd Annual Meeting of the Association for Computational Linguistics (Volume 1: Long Papers), 2025), as has providing detailed clinical guidelines to VLMs (Holland et al. "Specialized curricula for training vision language models in retinal image analysis." NPJ Digital Medicine 8.1, 2025).

- In many cases the authors evaluate both the image encoder and VLM in a few-shot testing scenario, arguing that this reflects “real-world medical applications where labeled data is scarce.” However, I am not sure whether fine-tuning with single-digit-number of images is representative of a realistic deployment scenario.

**Questions:**

- The authors only include very little technical details how they conducted the LoRA fine-tuning of MedGemma.

- I was confused that the option to output “not sure” appears in some of the depicted prompts (e.g. Figure 4) but not in others (e.g. Figure 3 or Table 3). I understand that space is limited in figures, but at the very least the authors should denote incompletely listed prompts with ellipses.

- The authors should provide more detail and context on the human evaluation study. Without any baseline method for comparison or even knowing the instructions to the rater and evaluation criteria, it is difficult to interpret the results provided in Figure 6.

- It appears that in Section 3.3 the authors make two contributions: using a pre-trained vision encoder to select the five most likely diagnoses as well as prompting the VLM to output a structured report. These contributions should be disentangled, and their benefit should be measured and reported individually.

- I had to read Table 2 several times to map the column names to the individual methods. I believe that the authors should select more expressive short descriptors for the different strategies.

- Additionally, I feel that the inclusion of the top-5 to top-1 strategy in Figure 1 is premature as it has not been introduced yet when the figure is referenced in the introduction. Especially since several of the conducted experiments are reported in multiple figures and tables.

- On a minor note, the order of the datasets in Figure 1 and the Tables differs.

---

> ### Author Response · Authors · 2025-11-25
> **Response to Reviewer hfSy [1/2]**
>
> We sincerely thank the reviewer for recognizing the significance of the research problem studied in the work, the effectiveness of the proposed strategies, and the comprehensiveness of our experiments. We also appreciate the positive remarks regarding the quality of our manuscript. We value the detailed suggestions provided, which have greatly enhanced the final version of our work!
>
> ## W1: Related Works on Prompting for VLMs
>
> We thank the reviewer for highlighting this relevant research. We agree that strategies like structured reporting [1] and providing clinical context [2] are established techniques relevant to our work. We will clarify their connections to this work in the revised manuscript.
>
> [1] Delbrouck et al. “Automated Structured Radiology Report Generation.” Proceedings of the 63rd Annual Meeting of the Association for Computational Linguistics (Volume 1: Long Papers), 2025
>
> [2] Holland et al. "Specialized curricula for training vision language models in retinal image analysis." NPJ Digital Medicine 8.1, 2025
>
>
> ## W2: Practicality of Few-shot Setting
>
> Thank you for this quesiton! We respectfully argue that this few-shot scenario is a highly realistic and critical for medical AI deployment. In many clinical applications, acquiring large, expert-labeled datasets for every new condition is infeasible. This is especially true for **rare diseases** (like ochronosis or the Neglected Tropical Diseases in the eSkinHealth dataset) or when deploying models in new, low-resource environments. Recent literature confirms that few-shot learning is indispensable for overcoming such **annotation scarcity** and **high labeling costs in medical imaging** [3, 4, 5]. The core promise of foundation models is their ability to adapt from very few examples. Our few-shot linear probe (LP) in Fig. 1 and Table 2  is used to benchmark the vision encoder's high capability. Our "Top-5 to Top-1" strategy is explicitly designed to be most effective in this low-data (1 to 8 shot) regime, where it consistently outperforms the linear probe alone. This makes it particularly valuable for the exact real-world challenges we aim to address.
>
> [3] Few-Shot Learning for Medical Image Classification, ICANN 2020
>
> [4] A systematic review of few-shot learning in medical imaging, Artificial Intelligence in Medicine 2024
>
> [5] Label-Efficient Deep Learning in Medical Image Analysis: Challenges and Future Directions, Arxiv 2025
>
> ## Q1: Technical Details on LoRA Fine-tuning
>
> Thank you for pointing this out! We adapted the fine-tuning code released by the MedGemma team. To ensure optimal performance, we conducted a grid search over key hyperparameters, exploring ranks of $\{4, 8, 16\}$, alphas of $\{8, 16, 32\}$, and learning rates in $\{5\text{e-}5, 1\text{e-}4, 2\text{e-}4\}$. We selected the configuration that maximized validation performance: **a rank of 8**, **alpha of 16**, and **learning rate of $1\text{e-}4$**. We trained for 10 epochs, applying LoRA to all linear layers in the model. We will add a dedicated subsection in the Appendix specifying these details and the full search space used for the results in Table 1.
>
>
> ## Q2: Clarifications on Abstention in Prompts
>
> This is a key point of clarification. The "NOT SURE" option was exclusively used for the "Abstention Test" (Fig. 4). The specific goal of that experiment was to test Hypothesis 2 by seeing if the model could abstain when presented with a masked image where no diagnosis was possible. For all other experiments, including the classification tasks in Fig. 3 and the ablation study prompts in Table 3, the task was standard forced-choice classification, so "NOT SURE" was not included in the options list. We will make this distinction explicit in the text. We also thank the reviewer for the suggestion and will add ellipses (...) to all truncated prompts in the figures to avoid confusion.

---

> ### Author Response · Authors · 2025-11-25
> **Response to Reviewer hfSy [2/2]**
>
> ## Q3: More Detail and Context on the Human Evaluation Study
>
> We appreciate the opportunity to clarify. This evaluation was designed to validate Hypothesis 3 that the VLM excels at description even when it fails at classification.
>
> - Instructions: The rater (a board-certified dermatologist) was given the 100 images and the corresponding generated text. The model was prompted to describe five specific criteria, as mentioned in the paper: [Location Site, Lesion Type, Shape/Border, Color, Texture]. An example of this prompt and a high-quality generated description is shown in Figure 7.
> - Evaluation Criteria: The score in Figure 6  directly reflects these instructions. As stated in the caption, the score [0-5] indicates "the number of these five clinical criteria correctly described". A score of 5 by the dermatologist means all five criteria were accurately and faithfully described.
> - Baseline: The "baseline" is not a competing description model, but rather the hallucinated, visually-ungrounded explanations the model produces in a standard classification prompt (our Fig. 3). The goal was to prove that the VLM's descriptive capabilities are strong (Fig. 6 & 7) in stark contrast to its poor zero-shot classification (Table 1).
>
> We will revise Section 3.3 to make these instructions and criteria more prominent.
>
>
> ## Q4: Disentangling Each Method's Contribution
>
> We thank the reviewer for this suggestion, and we are happy to confirm that this analysis is already present in our work. We have structured the paper to address this.
>
> - Section 3.1 introduces the "in-context" strategy.
> - Section 3.2 introduces the "describe-then-decide" strategy (which we believe the reviewer is referring to as "structured report").
> - Section 3.3 introduces the "Top-5 to Top-1" pipeline.
>
> Finally, Section 3.4 and Table 4 (Ablation study) explicitly disentangle these contributions and reports the individual performance benefit of each strategy ("in-context", "describe-first", "top-5 to top-1") before showing their cumulative effect in the "all combined" column . This table clearly shows the individual benefit of each component.
>
> ## Q5: Improve the Readability/Expressiveness of Table 2 Caption
>
> Thank you for the suggestion. We will revise the table caption and column headers for the final version, and improve the caption to ensure the comparison is clear to the readers.
>
> ## Q6: Inclusion of Methods before Introducing
>
> We apologize for the confusion, but we believe there may be a misunderstanding of Figure 1. **Figure 1 does not show our "Top-5 to Top-1" strategy.** The legend in Figure 1  shows Top-5 accuracy for linear probe and zero-shot VLM classification. The entire purpose of this figure, and its placement in the introduction, is to establish our core premise: the **massive performance gap** between the VLM's zero-shot performance (the VLM lines) and the underlying capability of its vision encoder (the LP lines) . This gap is what motivates our entire investigation (H1, H2, H3). Our "Top-5 to Top-1" solution is introduced much later, in Section 3.3, and its results are shown in Table 2 and Table 4. We will re-check the text in the introduction to ensure it references Figure 1 without ambiguity.
>
> ## Q7: Consistency of Order of the Datasets in Figures and Tables
>
> Thank you for catching this! We will standardize the order of datasets across all figures and tables in the final revision to improve consistency and readability.

---

### Author Response · Authors · 2025-12-04
**Summary of Rebuttal and Clarifications**

Dear Reviewers, ACs, SACs, and PCs,

We sincerely thank you for your dedication during this review process. We understand the workload involved in final decision-making. To assist in your assessment, we summarize the status of the discussion, highlighting the resolution of technical concerns and the extensive new experimental evidence added to the manuscript.

## 1. Consensus on Clinical Relevance

All three reviewers recognized the significance of the problem (over-reliance on language prior and under-reliance on vision branches in medical VLMs) and the rigorousness of our evaluation.

- **Reviewer hfSy** explicitly upheld their positive rating, stating: *"I am happy to uphold my positive overall rating, as I feel that the work is of interest to the growing community of scientists interested in applying VLMs for medical tasks."* (Posted by Reviewer hfSy below the response to Reviewer 6Nrz.)
- **Reviewer 6Nrz** acknowledged the work as a *"pioneer in this specific medical subdomain"* with a *"high level of completion."*
- **Reviewer bjJR**, though hasn't had the chance to post a follow-up comment, noted the significance of the issue we are studying and acknowledged the comprehensiveness of our evaluation.

## 2. Addressed Concern: Generalizability to Other VLMs for Dermatology (Reviewers bjJR, 6Nrz)

- **Concern:** Reviewers requested evidence that our findings were not specific to MedGemma.
- **Our Response (New Experiments):** We extended our full evaluation to **SkinVL (2025)**, a LLaVA-based SOTA model for dermatology with a distinct architecture from MedGemma.
  - **The Gap Exists:** We confirmed that SkinVL exhibits the same "paradox" where a linear probe on its vision encoder greatly outperforms the VLM fine-tuning and zero-shot (e.g., **50.16% vs 44.81 vs 12.77%** on Derm7pt).
  - **The Solution Works:** We applied our proposed inference pipeline to SkinVL. Despite its distinct architecture, our method yielded consistent gains (e.g., improving **from 10.05% to 63.91%** on Fitzpatrick17k), proving our approach is model-agnostic.
- **Status:** These results are now formalized in **Section 3.5 and Appendix D**, directly addressing the call for broader validation.

## 3. Addressed Concern: Evidence for Distribution Mismatch (Reviewer bjJR)

- **Concern:** The reviewer noted that evidence for Hypothesis 1 (Distribution Mismatch) was deductive.
- **Our Response (Direct Evidence):** We leveraged the open-source nature of SkinVL’s training data to provide direct empirical proof. We demonstrated a strong correlation between the classes most frequent in training (Lichen Planus, Pityriasis Rosea) and the model’s specific inference biases, conclusively validating that data imbalance is a root cause of the alignment failure.

## 4. Addressed Concern: Algorithmic Novelty vs. Practical Utility (Reviewer 6Nrz)

- **Concern:** Reviewer 6Nrz maintained a score of 4, arguing that the method is "superficial" compared to deep mechanistic interpretability or retraining.
- **Our Perspective for the AC:** We respectfully submit that this work targets **clinical translation**, not architectural re-design.
  - In resource-limited clinical settings, a solution that recovers significant performance (e.g., +20-30% accuracy) *without* the high cost of retraining or fine-tuning is a strategic advantage, not a weakness.
  - While we agree that solving modalities like Ultrasound is a worthy goal, we argue that Dermatology, with its high inter-class visual similarity, is a sufficiently complex and high-stakes domain to warrant this dedicated study. The fact that our method works effectively here is a valuable contribution to the medical AI community.

Conclusion: This paper provides the first systematic quantification of the vision-language performance gap in dermatology, validated across five datasets and two distinct, advanced VLMs (MedGemma and SkinVL). It further provides a targeted, fine-tuning-free inference pipeline to bridge this gap, enabling medical practitioners with limited data and computational resources to maximize the benefit of such foundation models. Given the rigorous analysis, comprehensive experiments, practical solution strategies, and new evidence addressing generalizability, we believe this work is a valuable contribution to reliable Medical AI.

Once again, we extend our sincere gratitude to all reviewers, ACs, SACs, and PCs for their significant efforts and contributions!

Best regards,

The Authors

---

### Meta-Review · Area_Chair_36kR · 2026-01-08

**Summary:**

This paper investigates the performance gap between vision encoders and full vision-language models (VLMs) in dermatological diagnosis, focusing on MedGemma-4B.

**Reviewer Concerns:**

Points of Agreement
All three reviewers recognized:
1. The clinical significance of the research problem
2. The comprehensive evaluation across multiple dermatological datasets
3. The rigor of the experimental methodology

Points of Divergence
1. Reviewer hfSy (Score: 6): Marginally positive, acknowledging the work's value to the medical VLM community despite modest algorithmic innovation.
2. Reviewer 6Nrz (Score: 4): Maintained reservations about (1) superficial methodology compared to deep mechanistic analysis, (2) limited generalizability beyond dermatology, particularly to challenging modalities like ultrasound, and (3) insufficient novelty for the broader VLM research community.
3. Reviewer bjJR (Score: 2): Most critical, raising concerns about (1) generalizability without multi-model evaluation, (2) missing statistical rigor, (3) circumstantial evidence for hypotheses, and (4) limited clinical utility assessment.

**Reviewer Scores:**

Strengths of Rebuttal

1. Generalizability validation: Added comprehensive evaluation on SkinVL (LLaVA-based architecture), demonstrating the phenomenon exists across different model families and that proposed methods generalize effectively
2. Direct empirical evidence: Leveraged SkinVL's public training data to directly validate Hypothesis 1 (distribution mismatch), showing correlation between training class frequency and inference bias
3. Methodological clarity: Clarified terminology (fine-tuning-free vs. training-free) and explained the strategic value of inference-time solutions
4. Statistical details: Committed to adding standard deviations and clarified experimental design choices

Remaining Limitations

1. Limited scope: Focus remains strictly on dermatology; generalization to other medical modalities (radiology, ultrasound) remains undemonstrated. Unclear whether description-based attention guidance scales to modalities with less natural-image-like characteristics
2. Limited algorithmic novelty: Proposed strategies (in-context learning, structured prompting, two-stage inference) combine existing techniques rather than introducing fundamentally new methods
3. Mechanistic depth: Does not provide token-level attention analysis or deep architectural insights into vision-language interaction
4. Clinical validation: Human evaluation limited to 100 images by single dermatologist; lacks inter-rater reliability and comprehensive clinical utility assessment

---

### Decision · Program_Chairs · 2026-01-26

Reject